# Kernel Space Conditional Distribution Alignment for Improving Group Fairness in Deepfake Detection

**Sayantan Das, Mojtaba Kolahdouzi, Ali Etemad**
*{sayantan.das, m.kolahdouzi, ali.etemad}@queensu.ca*
*Queen's University, Canada*

**Reviewed on OpenReview:** *https://openreview.net/forum?id=68Lv6v4N9J*

## Abstract

We introduce FairAlign, a new method to reduce bias and improve group fairness in deepfake detection by aligning conditional distributions of embeddings in a high-dimensional kernel space. Our approach reduces information related to sensitive attributes in the embedding space that could potentially bias the detection process, thus promoting fairness. FairAlign is a versatile plug-and-play loss term compatible with various deepfake detection networks and is capable of enhancing group fairness without compromising detection performance. In addition to applying FairAlign for reducing gender bias, we implement a systematic pipeline for the annotation of skin tones and promotion of fairness in deepfake detection related to this sensitive attribute. Finally, we perform the first comprehensive study toward quantifying and understanding the trade-off between fairness and accuracy in the context of deepfake detection. We use three public deepfake datasets FaceForensics++, CelebDF, and WildDeepfake to evaluate our method. Through various experiments, we observe that FairAlign outperforms other bias-mitigating methods across various deepfake detection backbones for both gender and skin tone, setting a new state-of-the-art. Moreover, our fairness-accuracy trade-off analysis demonstrates that our approach demonstrates the best overall performance when considering effectiveness in both deepfake detection and reducing bias. We release the code at: https://github.com/Mkolahdoozi/FairAlign.

## 1 Introduction

The ease of creating deepfakes necessitates the development of sophisticated methods for detection of content produced by *generative AI* models. To detect deepfakes, deep learning models have been recently utilized to discern subtle inconsistencies and artifacts common in synthetic content (Das et al., 2023; Wang et al., 2023; Cao et al., 2022). Although recent deepfake detectors achieve high detection rates, several studies (Trinh & Liu, 2021; Xu et al., 2022b; Nadimpalli & Rattani, 2022; Masood et al., 2022; Ju et al., 2024) have shown that detectors can exhibit bias toward specific groups with regards to sensitive attributes like gender, racial background, and others. For instance, it has been shown that certain state-of-the-art deepfake detectors output higher accuracies when deepfake content involve individuals with lighter skin tones (Hazirbas et al., 2021; Trinh & Liu, 2021). This can enable attackers to create malicious deepfakes aimed at more vulnerable groups. This highlights the need to consider group fairness as an equally important factor alongside detection accuracy in deepfake detectors.

We identify three key open problems with the current state of deepfake detection research: (**1**) Although a number of bias-mitigating strategies have been proposed for deepfake detection (Ju et al., 2024; Nadimpalli & Rattani, 2022), the embeddings generated by these detectors continue to retain information related to sensitive attributes, which could cause biases in detection of deepfake content. To demonstrate this empirically, we extract embeddings from the final layer of an EfficientNet-B4(Coccomini et al., 2022) detector trained with the DAG-FDD (Ju et al., 2024) bias mitigation method on the Celeb-DF dataset. We then project these embeddings into a 2D space using t-SNE and colored the resulting points based on the subjects' gender.

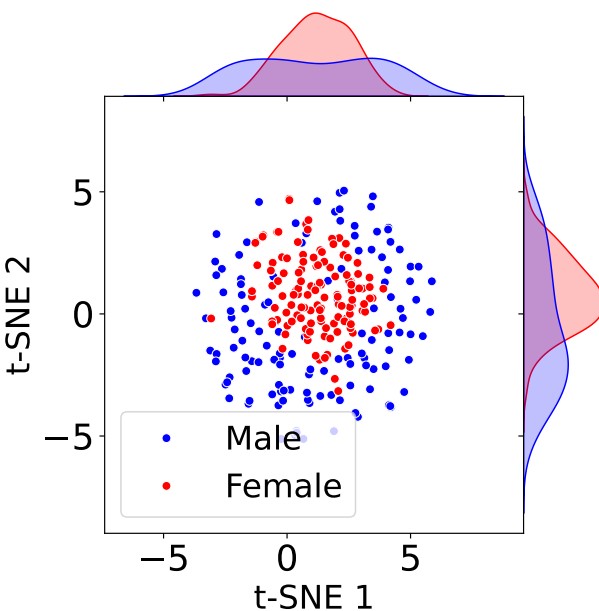

Figure 1: Representations for sample images from the Celeb-DF dataset obtained from the EfficientNet-B4 detector encoder (Coccomini et al., 2022) with DAG-FDD (Ju et al., 2024) bias-mitigation. The male/female clusters indicate the existence of gender-related information in the embeddings.

Figure 1 shows the result. It is evident that the embeddings corresponding to male and female subjects occupy different regions in the 2D space, demonstrating that gender-related information persists in the embeddings. As a result, we suggest that there is substantial room for improvement at the core of this problem. (**2**) Despite efforts to develop less biased deepfake detectors that work fairly across different populations, gender has been the main area of focus. While a few works have focused on 'ethnicity' as a second factor, we argue that 'skin tone' (Hazirbas et al., 2021) is an equally critical and, at the same time, more practical factor to focus on. We believe skin tone is possibly a different source for bias, since: (*i*) Each ethnicity might itself contain a variety of skin tones; (*ii*) Encoders are highly likely to discriminate based on color given its strong visual prominence; (*iii*) Ethnicity labels are prone to label noise (Heldreth et al., 2023). (**3**) Lastly, there exists a phenomenon referred to as the 'fairness-accuracy trade-off', which indicates that enhanced fairness may come at the cost of reduced accuracy (Little et al., 2022; Hazirbas et al., 2021). While some studies suggest variability in the existence of this trade-off (Maity et al., 2021; Wick et al., 2019; Dutta et al., 2020), the intertwined relationship of fairness and accuracy has been widely confirmed in prior works (Wang et al., 2021a; Li et al., 2021; Zhang et al., 2021). However, to our knowledge, this trade-off has not been studied in prior works in the context of deepfake detectors.

In this paper, to address the **first problem** mentioned above, we propose a novel loss term called *FairAlign*, for enhancing fairness via the alignment of conditional distributions of the embeddings in higher-dimensional kernel space. Our method aims to reduce the gap in detection performance across different sensitive attributes such as gender, thus mitigating the risk of biased outcomes. By leveraging the kernel space, our method integrates the cross-covariance and covariance operators of the conditional distributions of the embeddings given sensitive attributes obtained from deepfake detectors into the training process. Our method is a plug-and-play technique that can be integrated with other existing loss functions used in deepfake detection. Using our method, we perform multiple bias-mitigation experiments in deepfake detection on three public, real-world datasets (Celeb-DF (Li et al., 2020), FaceForensics++ (FF++) (Rossler et al., 2019), and WildDeepfake (Zi et al., 2020)), wherein we demonstrate the effectiveness of our method in improving the fairness of several state-of-the-art deepfake detectors while retaining strong detection performance. For the **second problem**, we propose a simple yet effective pipeline for detecting skin tones and using them to mitigate bias for this factor. Our method uses ArcFace (Deng et al., 2018) to detect and extract the face. Subsequently, we

select the facial skin regions using a pre-trained U-Net model (Xu et al., 2022a), based on which the average skin color is measured. Finally, we use the shortest Euclidean distance between the skin tone with respect to Monk Skin Tone (MST) (Heldreth et al., 2023) scale to determine the final skin tone. We then apply this pipeline to deepfake detection datasets for the first time, following which we perform skin tone bias mitigation experiments. We find that our method, FairAlign, is effective at reducing skin tone biases in deepfake detection datasets. Lastly, to address the **third problem**, we utilize two different metrics, Fairea (Hort et al., 2021) and Harmonic Mean (Lesota et al., 2022; Li et al., 2023), to combine both fairness and accuracy into unified indices, for the first time in the area of bias mitigation for deepfake detection. Our analysis shows that while some existing fairness-promoting techniques do indeed reduce bias to a good degree, this improvement comes at the cost of accuracy, hence not ideal for practical applications. The analysis further demonstrates that our proposed FairAlign maintains the highest performance in terms of both fairness and accuracy as per the unified metrics.

Our contributions are summarized as follows:

**(1)** For promoting group fairness in deepfake detection we propose a new loss term, FairAlign, that operates in the kernel space to reduce the distance between distributions of the representations learned by deepfake detectors given different sensitive attributes. Our method demonstrates effectiveness in improving the group fairness for state-of-the-art deepfake detectors while maintaining strong detection performance on three large-scale datasets, FF++ (Rossler et al., 2019), CelebDF (Li et al., 2020), and WildDeepfake (Zi et al., 2020).

**(2)** We analyze and improve fairness based on skin tones for deepfake detection tasks. We extract skin tones from existing deepfake datasets using the guidelines given by the MST scale(Heldreth et al., 2023), and apply our proposed FairAlign method for enhanced fairness. Our experiments demonstrate that FairAlign improves skin tone fairness across all state-of-the-art deepfake detectors. To our knowledge, this is the first attempt at reducing bias against different skin tones in the context of deepfake detection. Additionally, we enhance fairness based on the *intersection* of gender and skin tone in the context of deepfake detection for the first time.

**(3)** To objectively quantify the fairness-accuracy trade-off, we analyze two unified metrics for the first time in the realm of fair deepfake detection. Results show that our method is highly favorable as a bias-mitigating method that strikes a healthy balance between fairness and accuracy.

**(4)** To enable fast and reliable reproducibility, we release the code at: https://anonymous.4open.science/r/FairAlign-170F.

## 2 Related work

**Fairness in deepfake detection.** Prior works have demonstrated the existence of bias in deepfake detection tasks with respect to gender, age, and ethnicity (Hazirbas et al., 2021; Trinh & Liu, 2021; Pu et al., 2022; Xu et al., 2022b), while a few works have proposed solutions to mitigate this bias (Nadimpalli & Rattani, 2022; Ju et al., 2024). The work in (Trinh & Liu, 2021) evaluates bias in existing deepfake datasets and detection models for the first time in the literature; however, it doesn't take into account the intersectional bias. The evaluation of bias for a popular detection model (MesoInception-4) on FF++ dataset is done in (Pu et al., 2022). This work is limited in terms of the number of detection methods evaluated. A more comprehensive study is proposed in (Xu et al., 2022b), which evaluates fairness over three deepfake detection models. One recent work (Nadimpalli & Rattani, 2022) has attempted to mitigate the aforementioned biases through data-centric approaches, i.e., making the datasets like FF++ balanced with regards to different sensitive attributes like gender. The process of gender-balancing via data annotation is time-consuming and also showed limited improvement in fairness. The work in (Ju et al., 2024) applies conditional-value-at-risk loss to mitigate bias with regard to both gender and ethnicity in the context of deepfake detection. To our knowledge, this paper is the first and only one to directly provide a solution for bias in deepfake detection.

**Skin tone fairness.** While ethnicity has been considered for promoting fairness (Ju et al., 2024; Trinh & Liu, 2021; Xu et al., 2022b) in several deepfake detection literature, skin tone as a factor in reducing bias is more commonly addressed in areas like skin lesion classification (Kinyanjui et al., 2019; Heldreth et al., 2023).

To our knowledge, (Hazirbas et al., 2021) is the only prior work to study skin tone in the context of deepfake detection. In their study, researchers evaluate biases in deepfake detection by analyzing top models from the DeepFake Detection Challenge (DFDC) (Dolhansky et al., 2020) on the Casual Conversations dataset, which is rich in diversity across age, gender, and skin tone. Their analysis confirms the importance of skin tone as a crucial sensitive attribute for bias mitigation in deepfake detection.

**Fairness-accuracy trade-off.** While fairness-accuracy trade-off is a well-known phenomenon (Little et al., 2022; Dutta et al., 2020; Maity et al., 2021), only a handful of works have focused on introducing a quantitative measure to assess this trade-off (Dutta et al., 2020; Little et al., 2022; Wang et al., 2021b), although none are positioned in the context of deepfake detection. One such work defines the Fairness-Area-Under-the-Curve (FAUC) to empirically define the fairness-accuracy Pareto frontier (Little et al., 2022). FAUC provides a model-agnostic metric to measure the Pareto frontier. However, as mentioned in their work, FAUC is ineffective when intersectional fairness is involved or in cases where fairness and accuracy typically do not have an inversely proportional relationship. Another work (Wang et al., 2021b) approaches this trade-off through the lens of multi-task learning by proposing two metrics: Average Relative Fairness Gap and Average Relative Error. These metrics compare the Fairness-Performance Rate Gap and error rates of multi-task models to those of single-task models with the same architecture, providing a nuanced assessment of the balance between fairness and accuracy in multi-task learning. With a different perspective, (Dutta et al., 2020) approaches the trade-off between fairness and accuracy by quantifying separability with Chernoff information, challenging the use of biased datasets for performance measures, and advocating for ideal, unbiased datasets. Our work utilizes Fairea (Hort et al., 2021) and Harmonic Mean (Lesota et al., 2022), which were previously not explored in the context of deepfake detection. Fairea quantifies the trade-off by computing the area within an enclosed region formed by the baseline fairness values and the coordinates of any bias-mitigation method in a two-dimensional fairness-accuracy space. Harmonic Mean computes the trade-off using a straightforward formula that balances accuracy and fairness, ensuring neither is overlooked in the evaluation process.

## 3 Proposed method

We introduce a novel loss term called FairAlign for enhancing group fairness in deepfake detection. Training a deepfake detector using FairAlign causes the alignment of conditional distributions of embeddings given different sensitive attributes. Our fairness-enhancing approach is inspired by domain adaptation tasks (Mukherjee et al., 2022; Truong et al., 2023; Joshi & Burlina, 2021) and is considered an in-processing fairness technique (Han et al., 2024), as it intervenes directly within the learning algorithm to promote fairness.

**Problem setup.** Let us represent a triplet $\{(x_i, y_i, z_i)\}_{i=1}^N$, where $x_i$ denotes the embedding generated by deepfake detection model for the $i$th input image, $y_i$ is its corresponding forgery label (real, fake), and $z_i$ is the associated sensitive attribute (e.g., female, male, etc.). For simplicity, we can assume that the embeddings $x_i$ are independent and identically distributed. Additionally, let $X$, $Y$, and $Z$ be the random variables associated with the embeddings, their labels, and the sensitive attributes, respectively. Let the distribution of $X$ be defined over the set $\mathcal{X}$ and the distribution of $Z$ be defined over the set $\mathcal{Z}$. Also, let the cardinality of the set $\mathcal{Z}$ be equal to $\zeta$; for example, in the FF++ dataset, $\zeta = 2$ for gender, which corresponds to male and female. We alternatively denote embeddings $x_i$ as $x_i^a$ and $x_i^b$ where $a$ and $b$ denote elements in the set $\mathcal{Z}$. Let us denote $P_{(.)}$ as the probability distribution of an arbitrary random variable. Accordingly, the optimum condition for a fair classifier is denoted by

$$P_{\hat{Y}}(\hat{y}|Z = z_a) = P_{\hat{Y}}(\hat{y}|Z = z_b) \quad \forall z_a, z_b \in \mathcal{Z}, \tag{1}$$

where $\hat{Y}$ represents the random variable associated with the classifier's output, i.e., the predicted forgery category. This optimum condition is also known as demographic parity, which is the objective of many bias-mitigating methods (Srivastava et al., 2019). In this paper, we aim to reduce the information related to the sensitive attribute $z_i$ from the embeddings using our proposed loss.

**FairAlign.** Our overall goal is to accurately capture and minimize the distance between the distributions of embeddings given different sensitive attributes, through a novel loss term. The discrepancy between two distributions can be captured with regard to different statistical measures like expected value, covariance, etc. However, it has been shown that simply considering the arithmetic difference of such statistical measures,

particularly in lower dimensions, cannot effectively capture the discrepancy (Luo & Ren, 2021). As an example, Maximum Mean Discrepancy (MMD) (Gretton et al., 2012) uses the difference in expected values of two distributions in high dimensions (specifically, Reproducing Kernel Hilbert Space (RKHS) (Steinke & Schölkopf, 2008)) to render the discrepancy between two distributions. As another example, the Bures metric (Bhatia et al., 2017), which defines the discrepancy between two distributions as the difference between their covariance matrices, cannot effectively capture the discrepancy unless the data are mapped onto a high dimensional space. To avoid explicit data projection onto higher dimensions and the additional computational load required to measure the difference in high-dimensional space, kernel functions (Zhang et al., 2020) can be used to operate directly in low-dimensional space. Usage of kernel functions allows establishing the Bures metric in RKHS (Steinke & Schölkopf, 2008) where it is viable to be used as a distance metric (Zhang et al., 2020). This is also termed the Kernel Bures metric. Building on this concept, we utilize the Conditional Kernel Bures (CKB) metric, which is particularly designed for conditional distributions (Fukumizu et al., 2009; Luo & Ren, 2021). Another noteworthy point is that other domain adaptation methods like MMD are effective in aligning marginal discrepancy. Although effective, this can lead to discarding valuable discriminating information in label distributions (Luo & Ren, 2021). As CKB aligns conditional distributions, it preserves such information leading to higher classification accuracy.

As defined earlier, $\{x_i^a, z_i^a\}_{i=1}^n$ and $\{x_j^b, z_j^b\}_{j=1}^m$ are sets of embeddings corresponding to different sensitive attributes, drawn from the conditional distributions $P_{X|Z=z_a}$ and $P_{X|Z=z_b}$, respectively. Let us define kernel functions $k_{\mathcal{X}}$ and $k_{\mathcal{Z}}$ on the space of embeddings $\mathcal{X}$ and $\mathcal{Z}$, respectively. Further, we define $\phi(x) = k_{\mathcal{X}}(x, .)$ and $\psi(z) = k_{\mathcal{Z}}(z, .)$ as feature mappings from $\mathcal{X}$ to RKHS $\mathcal{H}_{\mathcal{X}}$ and $\mathcal{Z}$ to RKHS $\mathcal{H}_{\mathcal{Z}}$ respectively. Now, let us denote either $a$ or $b$ by the notation $a/b$. Accordingly $K_{XX}^{a/b}$, $K_{ZZ}^{a/b}$, and $K_{XX}^{ba}$ are the kernel matrices, where $(K_{XX}^{a/b})_{ij} = k_{\mathcal{X}}(x_i^{a/b}, x_j^{a/b})$, $(K_{ZZ}^{a/b})_{ij} = k_{\mathcal{Z}}(z_i^{a/b}, z_j^{a/b})$, and $(K_{XX}^{ba})_{ij} = k_{\mathcal{X}}(x_i^b, x_j^a)$.

The feature mappings can therefore be represented by $\Phi_{a/b} = \left[\phi(x_1^{a/b}), \ldots, \phi(x_{n/m}^{a/b})\right]$ and $\Psi_{a/b} = \left[\psi(z_1^{a/b}), \ldots, \psi(z_{n/m}^{a/b})\right]$. Consequently, as illustrated in (Luo & Ren, 2021), the empirical cross-covariance matrices are denoted by $\hat{A}_{XZ}^a = \frac{1}{n}\Phi_a J_n \Psi_a^\top$ and $\hat{A}_{XZ}^b = \frac{1}{m}\Phi_b J_m \Psi_b^\top$, with $J_n = (I_n - \frac{1}{n}\mathbb{1}_n\mathbb{1}_n^\top)$ as the centering matrix of size $n \times n$, $I_n$ as the identity matrix, and $\mathbb{1}_n$ as a vector of ones of dimension $n$ (similarly for $J_m$, $I_m$, and $\mathbb{1}_m$). The covariance matrices $\hat{A}_{XX}^{a/b}$ and $\hat{A}_{ZZ}^{a/b}$ are defined in a similar fashion. Moreover, the empirical conditional covariance is defined as:

$$\hat{A}_{XX|Z}^{a/b} = \hat{A}_{XX}^{a/b} - \hat{A}_{XZ}^{a/b}\left(\hat{A}_{ZZ}^{a/b} + \epsilon I\right)^{-1}\hat{A}_{ZX}^{a/b}, \tag{2}$$

where $\epsilon I$ acts as a regularizer to the $\hat{A}_{ZZ}$ matrix and $\epsilon > 0$ is the regularization factor. The regularization is done due to the rank deficiency of the matrix $\hat{A}_{ZZ}$. We denote the matrices

$$Q_a \triangleq I_n - \frac{1}{n\epsilon}\left[M_Z^a - M_Z^a\left(M_Z^a + \epsilon n I_n\right)^{-1}M_Z^a\right] \tag{3}$$

$$Q_b \triangleq I_m - \frac{1}{m\epsilon}\left[M_Z^b - M_Z^b\left(M_Z^b + \epsilon m I_m\right)^{-1}M_Z^b\right] \tag{4}$$

where the centralized kernel matrices are defined as: $M_Z^a = J_n K_{ZZ}^a J_n$ and $M_Z^b = J_m K_{ZZ}^b J_m$. Using the Cholesky decomposition (Schulman & Cramer, 1975) $Q_{a/b} = S_{a/b} S_{a/b}^T$, where $Q$ is a positive-definite matrix (Luo & Ren, 2021) and $S$ is the lower-triangular matrix obtained from the decomposition. Accordingly, we can reformulate the conditional covariance operator $\hat{A}_{XX|Z}^a$ as:

$$\hat{A}_{XX|Z}^a = \frac{1}{n}\Phi_a J_n S_a\left(\Phi_a J_n S_a\right)^T \tag{5}$$

and respectively for $\hat{A}_{XX|Z}^b$. The empirical CKB metric is accordingly defined as:

$$\begin{aligned}
\hat{d}_{CKB}^2(P_{X|Z=z_a}, P_{X|Z=z_b}) &= \hat{d}_{CKB}^2(\hat{A}_{XX|Z}^a, \hat{A}_{XX|Z}^b) \\
&= \epsilon\,\mathrm{tr}\left[M_X^a\left(\epsilon n\mathbf{I}_n + M_Z^a\right)^{-1}\right] + \epsilon\,\mathrm{tr}\left[M_X^b\left(\epsilon m\mathbf{I}_m + M_Z^b\right)^{-1}\right] - \frac{2}{\sqrt{m\times n}}\left\|\left(J_m S_b\right)^T K_{XX}^{ba}\left(J_n S_a\right)\right\|_*,
\end{aligned} \tag{6}$$

where $\| \cdot \|_*$ is the nuclear norm. The empirical CKB metric is differentiable and highly suitable for usage as a loss function.

**Total loss.** Based on Eq. (6), we define $\mathcal{L}_{FairAlign}$ as:

$$\mathcal{L}_{FairAlign} = \sum_{\forall (z_i, z_j) \in \mathcal{Z}} \hat{d}^2_{CKB}(P_{X|Z=z_i}, P_{X|Z=z_j}) \tag{7}$$

Additionally, we use a binary cross-entropy loss, $\mathcal{L}_{ce}$ for supervising the deepfake detector to discriminate between real and fake samples, defined as:

$$\mathcal{L}_{ce} = -\frac{1}{\Omega} \sum_{j=1}^{\Omega} [y_j \log(\hat{y}_j) + (1 - y_j) \log(1 - \hat{y}_j)] \tag{8}$$

Here, $\Omega$ denotes the total count of samples in the batch. Finally, we define the total loss as:

$$\mathcal{L} = \mathcal{L}_{ce} + \lambda \cdot \mathcal{L}_{\text{FairAlign}} \tag{9}$$

where $\lambda$ controls the contribution of the CKB term.

**Skin tone fairness enhancement.** As indicated earlier, in addition to a fair deepfake detection solution, we aim to detect skin tone. To this end, we first perform face detection using MobileFaceNet backbone (Chen et al., 2018) along with the ArcFace loss (Deng et al., 2018). We then employ the U-Net model presented in (Xu et al., 2022a) to segment skin regions from the extracted facial image. Next, we compute the average color of the facial skin pixels to obtain the overall skin tone. Finally, we intend to use a standard definition for characterizing the estimated skin tone. To do so, we use the MST scale (Heldreth et al., 2023). Therefore, to identify the corresponding tone from the MST scale, we compute the closest neighbour based on the Euclidean norm between the average tone and the tones in the MST scale. See Appendix for visual examples of skin tone bins along with the representative face crops for each.

**Fairness-accuracy trade-off assessment.** The final goal of our work is to conduct a comprehensive analysis of fairness-accuracy trade-off in the context of deepfake detection. Multiple studies have previously indicated the presence of an intrinsic trade-off between fairness and accuracy (Little et al., 2022; Hazirbas et al., 2021; Dutta et al., 2020; Wick et al., 2019), although not in the area of deepfake detection. To this end, we employ two metrics to characterize this trade-off in this context for the first time. (**1**) Fairea: The Fairea approach (Hort et al., 2021) first assesses how a model's predictions would change if it were less biased. This is done by manipulating the model's predictions to reflect a range of hypothetical scenarios from slightly to fully unbiased, which is referred to as 'mutation' in (Hort et al., 2021). The range of mutation can be from 10% to 100%, with 10% increments at each step. These adjusted predictions create a spectrum of potential fairness values within the model, referred to as the 'baseline'. This baseline, along with the coordinates of an arbitrary bias-mitigation method, form an enclosed region whose area quantifies the trade-off. When evaluating two bias mitigation methods, the one with the larger area is considered to have achieved a better fairness-accuracy trade-off. We illustrate this approach in Fig. 2. (**2**) Harmonic Mean: The Harmonic Mean (HM) takes into account both accuracy and fairness in the form of $HM = \frac{2A \times F}{A + F}$, where $A$ represents accuracy and $F$ stands for fairness. We apply the same rationale as various works that use F1 score (Lesota et al., 2022), wherein a harmonic mean formulation has been used to balance two diverging objectives (Li et al., 2023).

## 4 Experiment setup

**Datasets.** We conduct all the experiments based on three popular datasets, **FF++** (Rossler et al., 2019), **CelebDF** (Li et al., 2020), and **WildDeepfake** (Zi et al., 2020). FF++ comprises 1000 Baseline and 4000 forged videos with several visual quality levels, raw (no compression), high quality, and low quality. CelebDF contains 590 real and 5639 fake videos. WildDeepfake comprises *real-world* 7314 face sequences obtained from 707 deepfake videos. Since both the FF++ and CelebDF provide only the video-level labels, we sample frames out of these videos using FFMPEG (Tomar, 2006) and perform facial cropping on these frames using the ArcFace detection model (Deng et al., 2018).

Table 1: Results on FF++. Best results in each column are in **bold** and second-best results are underlined.

| Methods | Backbones | Gender $G_{FPR}$ | $F_{FPR}$ | $F_{EO}$ | Skin Tone $G_{FPR}$ | $F_{FPR}$ | $F_{EO}$ | Intersection $G_{FPR}$ | $F_{FPR}$ | $F_{EO}$ | Overall AUC↑ | FPR↓ | TPR↑ | ACC↑ |
|---|---|---|---|---|---|---|---|---|---|---|---|---|---|---|
| Baseline | EfficientNet-B3 | 1.97 | 1.97 | 8.15 | 11.05 | 10.86 | 32.19 | 14.38 | 22.65 | 44.13 | 94.72 | 20.25 | 97.21 | 94.09 |
| | RECCE | 1.27 | 1.27 | 9.14 | 18.81 | 29.65 | 25.07 | 30.26 | 69.38 | 82.34 | 98.05 | 21.20 | 98.21 | 94.74 |
| | EfficientNet-B4 | 1.97 | 1.97 | 7.85 | 11.56 | 10.86 | 34.12 | 23.89 | 20.56 | 42.13 | 95.91 | 20.25 | 97.21 | 94.09 |
| | MASDT | 1.38 | 1.38 | 19.71 | 14.64 | 11.89 | 11.39 | 18.07 | 14.32 | 41.46 | 96.21 | 3.65 | 97.13 | 97.60 |
| | AltFreezing | 2.82 | 2.82 | 10.54 | 18.37 | 9.85 | 18.09 | 12.02 | 33.74 | 40.74 | 97.84 | 8.42 | 96.27 | 98.10 |
| | *Average* | 1.88 | 1.88 | 11.08 | 14.89 | 14.62 | 24.17 | 19.72 | 32.13 | 50.16 | 96.55 | 14.75 | 97.21 | 95.72 |
| DRO$_{\chi^2}$ (Hashimoto et al., 2018) | EfficientNet-B3 | 0.23 | 0.23 | 4.42 | 4.71 | 6.58 | 12.38 | 6.30 | 12.32 | 42.85 | 94.37 | 8.06 | 89.60 | 89.66 |
| | RECCE | 0.33 | 0.33 | 5.46 | 6.15 | 9.08 | 11.71 | 20.27 | 24.97 | 64.89 | 98.32 | 7.99 | 96.48 | 95.98 |
| | EfficientNet-B4 | 0.54 | 0.54 | 3.64 | 11.40 | 17.00 | 19.89 | 15.03 | 35.28 | 51.11 | 93.90 | 1.96 | 98.10 | 96.77 |
| | MASDT | 0.64 | 0.64 | 15.68 | 6.51 | 9.04 | 7.24 | 12.93 | 13.04 | 41.15 | 98.29 | 3.22 | **98.96** | 97.37 |
| | AltFreezing | 2.67 | 2.67 | 9.83 | 5.10 | 9.20 | 17.36 | 11.55 | 31.60 | 26.19 | **98.86** | 8.03 | 97.22 | **98.75** |
| | *Average* | 0.88 | 0.88 | 7.81 | 6.77 | 10.18 | 13.72 | 13.22 | 23.44 | 45.24 | 96.75 | 5.85 | 96.07 | 95.71 |
| MMD (Deka & Sutherland, 2022) | EfficientNet-B3 | 0.35 | 0.35 | 6.65 | 5.88 | 4.78 | 12.91 | 5.62 | 12.84 | 44.95 | 93.57 | 8.49 | 94.11 | 94.01 |
| | RECCE | **0.16** | **0.16** | 6.19 | 7.14 | 10.18 | 9.19 | 18.98 | 20.08 | 66.46 | 96.96 | 8.42 | 93.65 | 93.53 |
| | EfficientNet-B4 | 0.39 | 0.39 | 2.59 | 8.72 | 13.03 | 15.05 | 10.34 | 25.47 | 39.50 | 92.64 | 1.58 | 97.08 | 95.60 |
| | MASDT | 0.32 | 0.32 | 14.20 | 5.89 | 6.94 | 5.87 | 10.20 | **11.43** | 29.18 | 97.04 | **1.23** | 98.02 | 98.32 |
| | AltFreezing | 2.02 | 2.02 | 7.06 | **3.74** | 7.18 | 12.61 | 7.78 | 22.30 | **21.01** | 97.82 | 5.78 | 96.14 | 98.05 |
| | *Average* | 0.65 | 0.65 | 7.34 | 6.27 | 8.42 | 11.13 | 10.58 | 18.42 | 40.22 | 95.61 | 5.10 | 95.80 | 95.90 |
| DAG-FDD (Ju et al., 2024) | EfficientNet-B3 | 0.67 | 0.67 | 5.36 | 11.48 | 9.58 | 13.50 | 12.87 | 19.34 | 46.08 | 97.01 | 8.40 | 92.87 | 92.65 |
| | RECCE | 0.75 | 0.75 | 5.71 | 14.68 | 19.41 | 19.33 | 25.40 | 38.17 | 76.24 | 98.33 | 12.01 | 96.80 | 95.23 |
| | EfficientNet-B4 | 0.61 | 0.61 | 4.86 | 13.81 | 16.85 | 20.55 | 17.50 | 30.88 | 52.63 | 94.15 | 21.58 | 95.60 | 92.92 |
| | MASDT | 0.58 | 0.58 | 18.70 | 10.60 | 9.84 | 7.36 | 16.04 | 14.27 | 30.39 | 96.95 | 5.67 | 97.63 | 98.29 |
| | AltFreezing | 2.64 | 2.64 | 10.01 | 11.18 | 9.27 | 17.42 | 11.30 | 32.62 | 37.69 | 97.10 | 7.87 | 95.59 | 97.44 |
| | *Average* | 1.05 | 1.05 | 8.93 | 12.35 | 12.99 | 15.63 | 16.62 | 27.06 | 48.61 | 96.71 | 11.11 | 95.70 | 95.31 |
| DAW-FDD (Ju et al., 2024) | EfficientNet-B3 | 0.34 | 0.34 | 6.53 | 6.79 | 11.67 | 12.63 | 8.43 | 12.57 | 43.72 | 95.96 | 8.22 | 91.43 | 91.49 |
| | RECCE | 0.45 | 0.45 | 7.95 | 6.99 | 9.96 | 13.95 | 23.54 | 25.44 | 54.95 | 98.35 | 8.15 | 94.59 | 94.10 |
| | EfficientNet-B4 | 0.55 | 0.55 | 3.71 | 13.65 | 17.35 | 20.30 | 15.34 | 36.00 | 56.19 | 90.44 | 2.00 | 96.91 | 95.40 |
| | MASDT | 0.45 | 0.45 | 17.71 | 8.62 | 8.86 | 9.87 | 12.05 | 13.29 | 33.42 | 96.86 | 5.41 | 97.81 | 98.13 |
| | AltFreezing | 2.72 | 2.72 | 10.03 | 6.20 | 9.39 | 17.71 | 11.79 | 32.24 | 37.54 | 97.64 | 8.19 | 95.91 | 97.81 |
| | *Average* | 0.90 | 0.90 | 9.19 | 8.45 | 11.45 | 14.89 | 14.23 | 23.90 | 45.16 | 95.85 | 6.39 | 95.33 | 95.39 |
| FairAlign (Ours) | EfficientNet-B3 | **0.16** | **0.16** | 5.78 | 3.97 | **3.59** | 10.15 | **4.74** | 13.09 | 45.74 | 92.87 | 8.59 | 95.19 | 93.66 |
| | RECCE | 0.19 | 0.19 | 4.98 | 6.02 | 10.03 | 10.50 | 14.21 | 21.58 | 57.54 | 96.74 | 8.53 | 93.60 | 93.03 |
| | EfficientNet-B4 | 0.39 | 0.39 | **2.07** | 6.54 | 12.29 | 11.79 | 8.10 | 22.95 | 45.08 | 91.78 | 1.47 | 97.24 | 95.75 |
| | MASDT | 0.29 | 0.29 | 12.00 | 4.01 | 5.28 | **5.38** | 8.26 | 11.72 | 23.07 | 97.23 | 1.67 | 98.11 | 98.46 |
| | AltFreezing | 1.74 | 1.74 | 6.03 | 4.10 | 7.05 | 10.17 | 5.90 | 23.78 | 31.23 | 97.97 | 5.36 | 96.26 | 98.13 |
| | *Average* | 0.55 | 0.55 | 6.17 | 4.93 | 7.65 | 9.60 | 8.24 | 18.62 | 40.53 | 95.32 | 5.12 | 96.08 | 95.81 |

**Evaluation metrics.** To assess fairness, we employ five bias metrics. First, following (Wang et al., 2022; Ju et al., 2024), we use the maximum difference in false positive rate (FPR) gap, equal FPR, and equal odds, denoted by $G_{FPR}$, $F_{FPR}$, and $F_{EO}$ respectively. These metrics are defined as:

$$G_{FPR} := \max_{\forall z_i, z_j \in \mathcal{Z}} \left| FPR_{z_i} - FPR_{z_j} \right| \tag{10}$$

$$F_{FPR} := \sum_{z_i \in \mathcal{Z}} \left| \frac{\sum_{i=1}^n \mathbb{1}_{[\hat{Y}_i=1, Z=z_i, Y_i=0]}}{\sum_{i=1}^n \mathbb{1}_{[Z=z_i, Y_i=0]}} - \frac{\sum_{i=1}^n \mathbb{1}_{[\hat{Y}_i=1, Y_i=0]}}{\sum_{i=1}^n \mathbb{1}_{[Y_i=0]}} \right| \tag{11}$$

$$F_{EO} := \sum_{z_i \in \mathcal{Z}} \sum_{q=0}^1 \left| \frac{\sum_{i=1}^n \mathbb{1}_{[\hat{Y}_i=1, Z=z_i, Y_i=q]}}{\sum_{i=1}^n \mathbb{1}_{[Z=z_i, Y_i=q]}} - \frac{\sum_{i=1}^n \mathbb{1}_{[\hat{Y}_i=1, Y_i=q]}}{\sum_{i=1}^n \mathbb{1}_{[Y_i=q]}} \right| \tag{12}$$

where $FPR_{z_i}$ represents the FPR scores of group $z_i$, $\mathbb{1}_{[]}$ denotes the indicator function, and $q$ represents the forgery label (1 is real and 0 is fake). Also note that in the special case when $Z$ corresponds to gender, i.e. $\zeta = 2$, metrics $G_{FPR}$ and $F_{FPR}$ return the same value. See the Appendix for the definition of demographic parity difference (DPD) and demographic parity ratio (DPR) fairness metrics. We report DPD and DPR results in the Appendix.

Finally, to assess the performance of different deepfake detectors, we utilize four widely-used metrics (Ju et al., 2024): the area under the curve (AUC), FPR, true positive rate (TPR), and top-1 accuracy (ACC).

**Baseline methods.** To validate the efficacy of our proposed loss term, $\mathcal{L}_{FairAlign}$, we integrate it into the training process of 5 state-of-the-art deepfake detector backbones: RECCE (Cao et al., 2022), MASDT (Das et al., 2023), AltFreezing (Wang et al., 2023), EfficientNet-B3 (Tan & Le, 2019), and EfficientNet-B4 (Coccomini et al., 2022). The objective is to assess the impact of $\mathcal{L}_{FairAlign}$ on the fairness of these models.

Table 2: Results on CelebDF. Best results in each column are in **bold** and second-best results are underlined.

| Methods | Backbones | Bias Metrics (%)↓ | | | | | | | | | Detection Metrics (%) | | | |
|---|---|---|---|---|---|---|---|---|---|---|---|---|---|---|
| | | Gender | | | Skin Tone | | | Intersection | | | Overall | | | |
| | | $G_{FPR}$ | $F_{FPR}$ | $F_{EO}$ | $G_{FPR}$ | $F_{FPR}$ | $F_{EO}$ | $G_{FPR}$ | $F_{FPR}$ | $F_{EO}$ | AUC↑ | FPR↓ | TPR↑ | ACC↑ |
| Baseline | EfficientNet-B3 | 3.82 | 3.82 | 7.54 | 11.37 | 9.85 | 14.09 | 21.25 | 45.47 | 70.74 | 94.13 | 9.25 | 92.27 | 95.04 |
| | RECCE | 2.71 | 2.71 | 3.14 | 18.81 | 27.65 | 30.07 | 30.26 | 67.38 | 80.34 | 94.05 | 12.20 | 95.21 | 95.74 |
| | EfficientNet-B4 | 1.21 | 1.21 | 5.15 | 10.05 | 20.86 | 34.12 | 13.38 | 22.65 | 40.13 | 93.91 | 10.25 | 97.21 | 93.09 |
| | MASDT | 0.48 | 0.48 | 3.71 | 8.64 | 10.89 | 9.39 | 12.07 | 15.32 | 16.46 | 96.61 | 5.05 | 94.13 | 95.60 |
| | AltFreezing | 1.71 | 1.71 | 8.52 | 7.25 | 12.63 | 22.65 | 23.83 | 42.65 | **40.13** | 95.91 | 8.25 | 94.11 | 96.09 |
| | *Average* | 1.99 | 1.99 | 5.61 | 11.22 | 16.38 | 22.06 | 20.16 | 38.69 | 49.56 | 94.92 | 9.00 | 94.59 | 95.11 |
| DRO$_{\chi^2}$ (Hashimoto et al., 2018) | EfficientNet-B3 | 3.40 | 3.40 | 6.83 | 8.10 | 9.20 | 12.36 | 11.55 | 39.60 | 66.95 | 93.63 | 6.32 | 93.22 | 96.62 |
| | RECCE | 0.84 | 0.84 | 4.66 | 6.85 | 7.80 | 10.71 | 13.27 | 22.97 | 71.89 | 91.32 | 10.99 | 93.48 | 90.98 |
| | EfficientNet-B4 | 0.44 | 0.44 | 3.64 | 12.40 | 17.00 | 19.89 | 15.03 | 35.28 | 53.11 | 90.90 | 11.96 | **98.10** | 94.77 |
| | MASDT | 0.22 | 0.22 | 3.68 | 8.51 | 10.70 | 9.24 | 9.93 | 15.04 | 16.15 | 95.29 | 5.54 | 93.96 | 96.37 |
| | AltFreezing | 0.33 | 0.33 | 5.23 | 6.16 | 10.83 | 17.91 | 11.14 | 32.32 | 42.85 | 94.37 | 6.62 | 90.01 | 95.66 |
| | *Average* | 1.05 | 1.05 | 4.81 | 8.40 | 11.11 | 14.02 | 12.18 | 29.04 | 50.19 | 93.10 | 8.29 | 93.75 | 94.88 |
| MMD (Deka & Sutherland, 2022) | EfficientNet-B3 | 3.02 | 3.02 | 5.06 | 6.74 | 11.18 | 10.61 | **7.78** | 28.30 | 51.17 | 94.20 | 6.88 | 92.14 | 95.25 |
| | RECCE | 0.76 | 0.76 | 4.89 | 7.14 | 8.18 | 12.19 | 13.98 | 24.08 | 64.46 | 92.96 | 10.42 | 92.65 | 91.53 |
| | EfficientNet-B4 | 0.29 | 0.29 | **2.59** | 8.72 | 13.03 | 15.05 | 10.34 | 25.47 | 37.50 | 92.64 | 12.58 | 97.08 | 94.60 |
| | MASDT | 0.31 | 0.31 | 3.20 | 6.89 | 7.94 | 7.87 | 9.20 | **12.43** | 14.18 | 96.04 | **4.40** | 92.02 | 97.32 |
| | AltFreezing | 0.52 | 0.52 | 4.54 | 6.85 | 10.97 | 18.83 | 11.32 | 28.84 | 44.95 | 94.57 | 6.91 | 96.15 | 97.01 |
| | *Average* | 0.98 | 0.98 | 4.06 | 7.27 | 10.26 | 12.91 | 10.52 | 23.82 | 42.45 | 94.08 | 8.24 | 94.01 | 95.14 |
| DAG-FDD (Ju et al., 2024) | EfficientNet-B3 | 3.65 | 3.65 | 8.01 | 9.18 | 9.27 | 11.42 | 12.30 | 42.62 | 67.96 | 93.02 | 10.72 | 92.59 | 95.20 |
| | RECCE | 1.45 | 1.45 | 3.71 | 12.68 | 17.41 | 19.33 | 15.40 | 36.17 | 64.24 | 94.33 | 10.01 | 95.80 | 93.23 |
| | EfficientNet-B4 | 0.61 | 0.61 | 4.86 | 12.81 | 16.85 | 20.55 | 17.50 | 30.88 | 52.63 | 92.15 | 11.58 | 95.60 | 94.92 |
| | MASDT | 0.58 | 0.58 | 3.70 | 8.60 | 10.84 | 9.36 | 12.04 | 15.27 | 16.39 | 95.95 | 5.55 | 93.63 | 96.29 |
| | AltFreezing | 0.97 | 0.97 | 6.63 | 7.81 | 11.83 | 18.24 | 20.81 | 39.34 | 46.08 | 97.20 | 7.02 | 94.75 | 96.65 |
| | *Average* | 1.45 | 1.45 | 5.38 | 10.22 | 13.24 | 15.78 | 15.61 | 32.86 | 49.46 | 94.53 | 8.98 | 94.47 | 95.26 |
| DAW-FDD (Ju et al., 2024) | EfficientNet-B3 | 3.56 | 3.56 | 7.03 | 7.20 | 6.39 | 9.18 | 11.79 | 42.24 | 67.45 | 93.41 | 9.98 | 92.91 | 95.17 |
| | RECCE | 0.95 | 0.95 | 4.75 | 6.99 | 7.96 | 11.95 | 13.54 | 23.44 | 62.95 | 92.35 | 12.15 | 94.59 | 92.10 |
| | EfficientNet-B4 | 0.45 | 0.45 | 3.71 | 12.65 | 17.35 | 20.30 | 15.34 | 36.00 | 54.19 | 91.44 | 12.00 | 96.91 | 93.40 |
| | MASDT | 0.38 | 0.38 | 3.71 | 8.62 | 10.86 | 9.37 | 12.05 | 15.29 | 16.02 | 95.56 | 4.55 | 94.81 | 96.13 |
| | AltFreezing | 0.43 | 0.43 | 5.34 | 5.96 | 10.62 | 17.82 | 11.53 | 37.57 | 43.72 | 96.30 | 7.26 | 93.32 | 97.49 |
| | *Average* | 1.15 | 1.15 | 4.91 | 8.28 | 10.64 | 13.72 | 12.85 | 30.91 | 48.87 | 93.81 | 9.19 | 94.51 | 94.86 |
| FairAlign (Ours) | EfficientNet-B3 | 2.67 | 2.67 | 4.39 | 5.19 | **5.54** | 7.77 | 9.08 | 23.78 | 41.39 | 94.71 | 5.61 | 93.26 | 96.33 |
| | RECCE | 0.86 | 0.86 | 4.98 | 7.32 | 8.30 | 10.50 | 14.21 | 24.58 | 55.54 | 93.74 | 7.53 | 91.60 | 94.03 |
| | EfficientNet-B4 | 0.29 | 0.29 | 2.77 | 6.54 | 12.99 | 11.79 | 12.10 | 22.95 | 33.08 | 93.78 | 11.47 | 97.24 | 94.75 |
| | MASDT | **0.19** | **0.19** | 3.00 | **5.01** | 7.88 | **7.38** | 9.16 | 12.72 | **12.07** | 96.23 | 5.40 | 92.11 | 97.46 |
| | AltFreezing | 0.29 | 0.29 | 4.86 | 5.78 | 10.17 | 16.95 | 11.45 | 31.09 | 45.74 | **97.87** | 6.28 | 96.94 | **97.66** |
| | *Average* | 0.86 | 0.86 | 4.00 | 5.97 | 8.98 | 10.88 | 11.20 | 23.02 | 37.56 | 95.27 | 7.26 | 94.23 | 96.05 |

Table 3: Bias for ethnicity vs. skin tone.

| Methods | BB | Ethnicity | | | Skin tone | | |
|---|---|---|---|---|---|---|---|
| | | $G_{FPR}$ | $F_{FPR}$ | $F_{EO}$ | $G_{FPR}$ | $F_{FPR}$ | $F_{EO}$ |
| Baseline | MASDT | 10.56 | 12.99 | 11.18 | 14.64 | 11.89 | 11.39 |
| | AltFreezing | 8.11 | 7.21 | 21.48 | 18.37 | 9.85 | 18.09 |

For a thorough analysis, we benchmark our approach against 4 state-of-the-art bias-mitigating methods: **DAG-FDD** (Ju et al., 2024), **DAW-FDD** (Ju et al., 2024), **DRO-$\chi^2$** (Chai et al., 2022), and **MMD** loss (Deka & Sutherland, 2022). Furthermore, to establish baseline performances, each model is also trained without any fairness-enhancing module.

**Implementation details.** All experiments are conducted using the PyTorch framework (Paszke et al., 2019) on up to 8 NVIDIA A100 PCIE GPU cards. We train all methods using the AdamW optimizer (Loshchilov & Hutter, 2017) with a batch size of 32, a maximum of 100 epochs, and a learning rate of 0.001. The optimizer employs first and second momentum decays of 0.9 and 0.999, respectively. Additionally, we use a weight decay of 0.01 to refine the training process. The learning rate is adjusted using a step scheduler, which decreases the learning rate by a factor of 0.5 every 5 epochs. The video frame input size is set to 380 pixels, with training augmentations including resizing, normalization and horizontal flipping. For the face detection process mentioned in Section 3, we use the InsightFace toolkit (Deng et al., 2022). For the $\epsilon$ used in Equation Eq. (2), we adhere to 0.01 as per the design choice outlined in (Luo & Ren, 2021). Furthermore, we employ Gaussian kernel for both the embedding space and the sensitive attribute space. The sigma parameter in the Gaussian kernel is dynamically set to the mean of the pairwise squared Euclidean distances for each mini-batch.

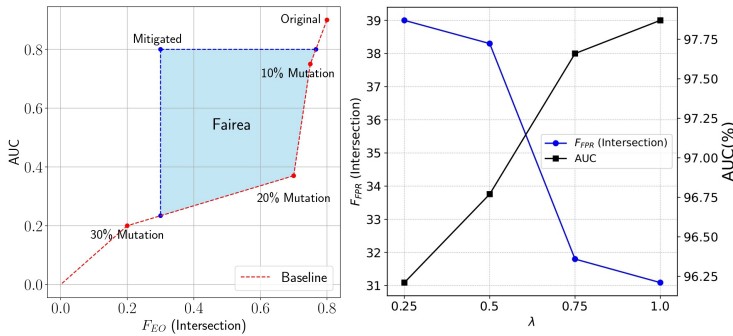

Figure 2: Left: schematic of the Fairea (Hort et al., 2021). Right: effect of $\lambda$ on bias mitigation.

Table 4: Cross-dataset results.

| Methods | BB | $\mathbf{G_{FPR}}$ | $\mathbf{F_{EO}}$ | AUC | Acc |
|---------|-----|------|-------|-------|-------|
| Baseline | MASDT | 1.55 | 10.51 | 80.21 | 88.76 |
|  | AltFreezing | 4.11 | 18.67 | 92.61 | 95.24 |
| Ours | MASDT | 0.78 | 6.34 | 82.78 | 91.34 |
|  | AltFreezing | 0.91 | 6.78 | 94.35 | 95.87 |

## 5  Results

**Performance.** We present a thorough analysis of the performance of our approach on the FF++ and CelebDF datasets in Tables 1 and 2 respectively. Additionally, we present the results of our method on the WildDeepfake dataset in the Appendix. First, evaluating the average results of FairAlign against the baselines demonstrates that our approach substantially promotes fairness across all three metrics for both gender and skin tone, as well as the intersection of the two. A similar trend is observed for the additional two metrics (DPD and DPR) presented in the Appendix. Comparing the performance of our method averaged across backbones, against other fairness-promoting solutions, we observe that our method generally achieves the best fairness scores, with exceptions in the intersection group, where MMD achieves marginally higher scores for $F_{FPR}$ and $F_{EO}$ on the FF++ dataset. Similarly, MMD obtains slightly better results in the intersection group with $G_{FPR}$ on CelebDF. While our method achieves the highest detection AUC on the CelebDF dataset with the AltFreezing backbone, it exhibits a slightly lower AUC compared to the baseline on the FF++ dataset. In the next subsection, we will analyze this outcome in detail and demonstrate that, considering the trade-off between detection and fairness, our method remains the strongest.

Delving deeper into the results, we observe that on FF++, FairAlign applied to EfficientNet-B3 achieves the lowest $G_{FPR}$ (and similarly $F_{FPR}$) of 0.16% in gender, in a tie with MMD when applied to the RECCE network. Similarly in CelebDF, FairAlign with MASDT achieves the lowest $G_{FPR}$ (and $F_{FPR}$) in gender with a score of 0.19%. For $F_{EO}$ our method obtains the best performance on gender when coupled with EfficientNet-B4 on FF++, while on CelebDF, MMD results in the best outcome with the same backbone. For skin tone, on FF++, MMD achieves the best performance for $G_{FPR}$ along with the AltFreezing deepfake detection method. However, for $F_{FPR}$ and $F_{EO}$, our method obtains the best results with EfficientNet-B3 and MASDT backbones respectively. On CelebDF, our method consistently achieves the lowest bias when coupled with MASDT, EfficientNet-B3, and MASDT, for the three metrics respectively. For the intersection of the two

Table 5: Fairness-accuracy trade-off on the FF++ and CelebDF datasets.

|  | Eff.-B3 | | | | RECCE | | | | Eff.-B4 | | | | MASDT | | | | AltFreezing | | | |
|---|---|---|---|---|---|---|---|---|---|---|---|---|---|---|---|---|---|---|---|---|
|  | FF++ | | CelebDF | | FF++ | | CelebDF | | FF++ | | CelebDF | | FF++ | | CelebDF | | FF++ | | CelebDF | |
|  | Fairea↑ | HM↑ | Fairea↑ | HM↑ | Fairea↑ | HM↑ | Fairea↑ | HM↑ | Fairea↑ | HM↑ | Fairea↑ | HM↑ | Fairea↑ | HM↑ | Fairea↑ | HM↑ | Fairea↑ | HM↑ | Fairea↑ | HM↑ |
| DRO$^2_\chi$ (Hashimoto et al., 2018) | 0.06 | 1.34 | 0.02 | 1.15 | **0.06** | 1.20 | 0.02 | 1.10 | 0.04 | 1.26 | 0.02 | 1.22 | 0.04 | 1.39 | 0.03 | 1.65 | 0.05 | 1.57 | 0.03 | 1.34 |
| MMD (Deka & Sutherland, 2022) | 0.05 | 1.31 | 0.03 | 1.27 | 0.05 | 1.17 | 0.03 | 1.16 | **0.06** | **1.35** | 0.03 | 1.36 | 0.04 | 1.51 | 0.03 | 1.69 | 0.03 | **1.62** | 0.03 | 1.32 |
| DAG-FDD (Ju et al., 2024) | 0.04 | 1.34 | 0.03 | 1.13 | 0.04 | 1.12 | 0.02 | 1.17 | 0.05 | 1.25 | 0.02 | 1.24 | 0.04 | 1.49 | 0.03 | 1.65 | 0.04 | 1.42 | 0.03 | 1.34 |
| DAW-FDD (Ju et al., 2024) | 0.05 | **1.35** | 0.02 | 1.14 | **0.06** | **1.27** | 0.03 | 1.16 | 0.05 | 1.19 | **0.04** | 1.22 | **0.05** | 1.46 | **0.04** | 1.65 | 0.03 | 1.42 | 0.03 | 1.35 |
| FairAlign (Ours) | **0.07** | 1.31 | **0.04** | **1.36** | 0.05 | 1.24 | **0.04** | **1.23** | 0.04 | 1.30 | 0.03 | **1.43** | **0.05** | **1.59** | **0.04** | **1.72** | **0.06** | 1.50 | **0.04** | **1.36** |

Table 6: Fairness metrics for Gaussian vs. linear kernel.

| Backbone | Gender | | | Skin tone | | |
|---|---|---|---|---|---|---|
| | $G_{FPR}$ | $F_{FPR}$ | $F_{EO}$ | $G_{FPR}$ | $F_{FPR}$ | $F_{EO}$ |
| **FairAlign (Gaussian Kernel)** | | | | | | |
| EfficientNet-B3 | 0.16 | 0.16 | 5.78 | 3.97 | 3.59 | 10.15 |
| EfficientNet-B4 | 0.39 | 0.39 | 2.07 | 6.54 | 12.29 | 11.79 |
| **Linear Kernel** | | | | | | |
| EfficientNet-B3 | 0.46 | 0.46 | 6.69 | 6.18 | 5.81 | 13.51 |
| EfficientNet-B4 | 0.48 | 0.48 | 2.75 | 10.42 | 13.78 | 16.64 |

(gender and skin tone), on FF++, our approach outperforms the others based on $G_{\mathrm{FPR}}$ using EfficientNet-B3, while MMD shows better performance on $F_{\mathrm{FPR}}$ and $F_{\mathrm{EO}}$ using MASDT and AltFreezing respectively. On CelebDF, MMD achieves better intersection results based on $G_{\mathrm{FPR}}$ and $F_{\mathrm{FPR}}$ using EfficientNet-B3 and MASDT, while ours outperforms others based on $F_{\mathrm{EO}}$ using the MASDT method.

Comparing the bias metrics for gender with those of skin tone, we notice considerably higher, i.e., more biased, values for skin tone. We believe this due to two main reasons. First, gender is currently defined as a binary class in the dataset, whereas our definition of skin tones consist of 10 unique classes. This difference between the number of classes is an important reason behind skin tone showing more bias as measured by the metrics. The second reason could be that skin tone is inherently more challenging in terms of bias mitigation, for instance due to the heavily imbalanced nature of the datasets in this regard. We present the distributions for gender and measured skin tones in the Appendix, where we observe a less balanced, i.e., long-tailed distribution for skin tones. To compare the bias related to skin color and ethnicity, we present the fairness metrics for both in Table 3 for the MASDT and AltFreezing backbones. As demonstrated, there is a significant difference in the bias associated with ethnicity compared to that associated with skin tone. For instance, the AltFreezing backbone achieves a $G_{FPR}$ of 8.11 for ethnicity, whereas it achieves a $G_{FPR}$ of 18.37 for skin tone. This indicates that skin tone is a distinct/complementary source of bias, introducing more severe bias into the system. We also test the performance of FairAlign on ethnicity sensitive attribute. The results are presented in Appendix.

Our method demonstrates strong deepfake detection performances across all four metrics for FF++ and CelebDF datasets. On the FF++ dataset, FairAlign obtains very competitive results, while on CelebDF, it generally achieves better performances with respect to others. A detailed comparison between both performance aspects (fairness alongside deepfake detection) highlights that drawing a high-level conclusion about the best fairness promoting approach remains complicated and nuanced when considering the deepfake detection results. This is especially the case on FF++ where the trade-off between fairness and performance seems more complex. The next subsection discusses this phenomenon.

To investigate the generalization and fairness of the proposed method on unseen data, we conduct cross-dataset experiments. We train MASDT and AltFreezing on FF++, both without any fairness-enhancing loss as a baseline and with the proposed loss. We then evaluate their performances on CelebDF. The results are presented in Table 4, where we observe that for deepfake detection with MASDT, our proposed method outperforms the baseline by nearly 3% in accuracy and over 2% in AUC, indicating that our loss can help improve generalization on unseen data while mitigating bias. Furthermore, the proposed loss achieves significantly lower $F_{EO}$ and $G_{FPR}$ compared to the baseline, highlighting enhanced group fairness. Similar results are observed for the AltFreezing backbone, as presented in the table.

Finally, we evaluate the computational overhead of our method on CelebDF. First we observe that at test time, our method does not add any overhead given that the loss is only measured during training. Next, when evaluating the overhead during trainig, we observe that our method averages 167 seconds per epoch, compared to 187 seconds for (Hashimoto et al., 2018) and 142 seconds for (Deka & Sutherland, 2022).

**Trade-off analysis.** To perform an analysis on fairness-accuracy trade-off, we use $1/F_{\mathrm{EO}}$ following (Ju et al., 2024) to represent fairness performance for the intersection of gender and skin tone to capture a holistic view of both sensitive attributes. Moreover, following (Wang et al., 2023) we select AUC to represent the

Table 7: Fairness metrics for gender sensitive attribute in 3 different scenarios: 1. Baseline: no fairness enhancing method is applied 2. FairAlign with gender as the sensitive attribute 3. FairAlign with skin tone as the sensitive attribute.

| Backbone | Gender | | |
|---|---|---|---|
| | $G_{FPR}$ | $F_{FPR}$ | $F_{EO}$ |
| **Baseline** | | | |
| **MASDT** | 1.38 | 1.38 | 19.71 |
| **AltFreezing** | 2.82 | 2.82 | 10.54 |
| **FairAlign (Targeting Gender)** | | | |
| **MASDT** | 0.29 | 0.29 | 12.00 |
| **AltFreezing** | 1.74 | 1.74 | 6.03 |
| **FairAlign (Targeting Skin Tone)** | | | |
| **MASDT** | 0.87 | 0.87 | 18.79 |
| **AltFreezing** | 2.80 | 2.80 | 10.13 |

the deepfake detection performance of different methods. Using these metrics, we present Fairea and HM as discussed in Section 3, and present the results in Table 6. We observe that most of the time FairAlign outperforms other bias-mitigation methods across different backbones when considering both fairness and accuracy. For instance, FairAlign achieves the highest Fairea and HM scores on CelebDF for MASDT backbone. On the FF++ dataset, the best Fairea score is obtained by our method for the Efficient-B3 backbone. An important observation from this analysis is that while theoretically both metrics (Fairea and HM) are capable of quantifying the fairness-accuracy trade-off, Fairea seems to produce less discriminatory outcomes. In contrast, HM generates a wider range of values, offering a more effective and discriminative means of capturing the trade-off.

**Gaussian kernel effectiveness.** To validate the effectiveness of employing a Gaussian kernel, we conduct an experiments. In this experiment, we compare our proposed FairAlign with a Gaussian kernel against a variant using a standard linear kernel ($\mathcal{K}_{linear}(x_i, x_j) = x_i^t x_j$). We use FF++ dataset, treating gender and skin tone as the sensitive attributes. The results are presented in Table 6. The results demonstrate a clear and consistent trend: the Gaussian kernel outperforms the linear kernel in reducing bias across both backbones and sensitive attributes. For instance, with the EfficientNet-B3 backbone and gender as the sensitive attribute, our proposed method with the Gaussian kernel achieves a $G_{FPR}$ of 0.16, a nearly three-times improvement over the linear kernel's score of 0.46.

**Cross-Attribute impact.** Existing work show that mitigating bias with regards to one sensitive attribute may exacerbate another. To directly investigate this in the case of FairAlign, we conduct an experiments to analyze the cross-attribute impact of FairAlign. Specifically, on the FF++ dataset with the MASDT and AltFreezing backbones, we trained three separate models: 1. Baseline with no fairness-enhancing method. 2. A model where FairAlign is applied to mitigate gender bias directly. 3. A model where FairAlign is applied to mitigate skin tone bias. For all the models described, we report fairness metrics for gender as the sensitive attribute in Table 7. The results show that even when FairAlign directly targets only skin tone, the gender bias is still reduced across all metrics compared to the baseline. For example, on AltFreezing, $F_{EO}$ improves from 10.54 to 10.13, and on MASDT, $G_{FPR}$ improves from 1.38 to 0.87. This demonstrates that FairAlign does not exacerbate the non-target bias beyond the baseline. Instead, it yields a positive, albeit smaller, fairness improvement on non-target attributes.

**Impact of $\lambda$ on fairness.** We further investigate the impact of the $\lambda$ hyperparameter in Eq. (9) on the bias metrics. Fig. 2 illustrates the relationship between $F_{\text{FPR}}$ (Intersection), AUC, and various values of $\lambda$ using the AltFreezing backbone on the CelebDF dataset. The figure demonstrates a significant reduction in bias, especially between $\lambda = 0.50$ and $\lambda = 1.0$, and an increase in the AUC. We illustrate the impact of $\lambda$ for the remainder of the bias metrics in the Appendix. According to these experiments, we set $\lambda = 1.0$ throughout all the experiments.

# 6 Conclusion

We introduced FairAlign, a novel loss term aimed at enhancing group fairness in deepfake detection by aligning conditional embedding distributions in a high-dimensional kernel space. Additionally, we consider skin tone as an important factor toward bias in deepfake detection for the first time. Our method shows state-of-the-art or competitive performance in mitigating bias, while maintaining a strong balance when considering the fairness-accuracy trade-off.

**Acknowledgments**

This work was partially funded by Irdeto Canada Corporation and the Natural Sciences and Engineering Research Council of Canada (NSERC), and was enabled in part by the support provided by the Digital Research Alliance of Canada.

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

## A  Appendix

### A.1  Skin Tone Scale with Representative Face Crops

We used Monk skin tone scale to compute the skin color of the cropped faces. To visually demonstrate the results, we add the panel of the 10-level Monk skin tone scale with representative face crops for each bin. Figure 3 shows the result. This figure shows that within each skin tone, there exist several ethnicity groups. This is in line with the previous finding in the literature that skin tone and ethnicity are 2 different concepts. We analyzed and mitigated the bias for both ethnicity and skin color sensitive attributes in this paper.

### A.2  Fairness Metric Definition

Following (Weerts et al., 2023), we use demographic parity difference (DPD) and demographic parity ratio (DPR) as fairness metrics, formulated as:

$$\text{DPD} = \max_{z_i \in \mathcal{Z}} P(\hat{Y} = 1|Z = z_i) - \min_{z_i \in \mathcal{Z}} P(\hat{Y} = 1|Z = z_i) \tag{13}$$

$$\text{DPR} = \frac{\min_{z_i \in \mathcal{Z}} P(\hat{Y} = 1|Z = z_i)}{\max_{z_i \in \mathcal{Z}} P(\hat{Y} = 1|Z = z_i)}. \tag{14}$$

DPD measures the disparity in positive outcomes across different groups, with an ideal value of 0 indicating no disparity. Conversely, DPR assesses the relative disparity, with an ideal value of 1 suggesting equal positive outcome rates across all groups.

### A.3  Additional Results

### A.3.1  Metrics

In addition to utilizing bias metrics $G_{\text{FPR}}$, $F_{\text{FPR}}$, and $F_{\text{EO}}$ in our experiments, we also employ DPR and DPD metrics and present the results in Tables 8 and 9. We observe that the findings align with those reported in the main paper. Averaging across different backbones, our method records the lowest DPD and highest DPR for all groups categorized by gender, skin tone, and intersectional attributes. Among the backbones, AltFreezing detector consistently ranks as the best or second-best for gender metrics across both FF++ and CelebDF benchmarks. Similarly, EfficientNet-B3 frequently secures the top scores for metrics related to skin tone and intersectional groups in both datasets.

### A.3.2  Performance on WildDeepfake Dataset

We report the performance of our proposed method on the WildDeepFake dataset in Table 10. For the gender sensitive attribute and its intersection with skin tone, FairAlign outperforms both DAG-FDD and the Baseline

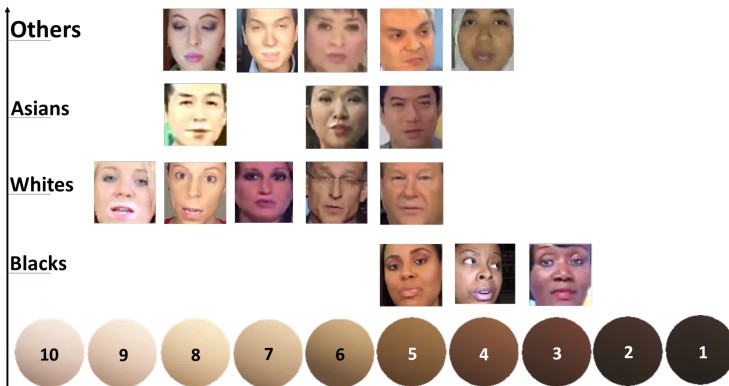

Figure 3: Visual example of Monk skin tone scale along with representative face images.

Table 8: DPR and DPD results on FF++. Best results in each column are in **bold** and second-best results are underlined.

| Methods | Backbones | Bias Metrics | | | | | |
| | | Gender | | Skin Tone | | Intersection | |
| | | DPD ↓ | DPR ↑ | DPD ↓ | DPR ↑ | DPD ↓ | DPR ↑ |
|---|---|---|---|---|---|---|---|
| Baseline | EfficientNet-B3 | 0.29 | 0.79 | 0.38 | 0.64 | 0.55 | 0.56 |
| | RECCE | 0.39 | 0.77 | 0.39 | 0.67 | 0.49 | 0.72 |
| | EfficientNet-B4 | 0.49 | 0.69 | 0.46 | 0.73 | 0.53 | 0.66 |
| | MASDT | 0.47 | 0.77 | 0.54 | 0.68 | 0.42 | 0.52 |
| | AltFreezing | 0.38 | 0.67 | 0.47 | 0.73 | 0.54 | 0.65 |
| | *Average* | 0.40 | 0.74 | 0.45 | 0.69 | 0.51 | 0.62 |
| $DRO_{\chi^2}$ (Hashimoto et al., 2018) | EfficientNet-B3 | 0.29 | 0.82 | 0.29 | 0.85 | 0.40 | 0.87 |
| | RECCE | 0.13 | 0.89 | 0.20 | 0.88 | 0.28 | 0.83 |
| | EfficientNet-B4 | 0.39 | 0.70 | 0.36 | 0.84 | 0.37 | 0.87 |
| | MASDT | 0.17 | 0.88 | 0.34 | 0.79 | 0.35 | 0.63 |
| | AltFreezing | 0.18 | 0.88 | 0.27 | 0.84 | 0.30 | 0.76 |
| | *Average* | 0.23 | 0.83 | 0.29 | 0.84 | 0.34 | 0.79 |
| MMD (Deka & Sutherland, 2022) | EfficientNet-B3 | 0.20 | 0.85 | 0.19 | 0.87 | 0.35 | 0.89 |
| | RECCE | 0.16 | 0.90 | 0.31 | 0.89 | 0.26 | 0.86 |
| | EfficientNet-B4 | 0.40 | 0.61 | 0.47 | 0.85 | 0.44 | 0.89 |
| | MASDT | 0.28 | 0.90 | 0.48 | 0.81 | 0.27 | 0.74 |
| | AltFreezing | 0.14 | 0.90 | 0.28 | 0.85 | 0.24 | 0.83 |
| | *Average* | 0.24 | 0.83 | 0.35 | 0.85 | 0.31 | 0.84 |
| DAG-FDD (Ju et al., 2024) | EfficientNet-B3 | 0.20 | 0.82 | 0.22 | 0.85 | 0.32 | 0.88 |
| | RECCE | 0.22 | 0.80 | 0.41 | 0.89 | 0.36 | 0.85 |
| | EfficientNet-B4 | 0.30 | 0.72 | 0.27 | 0.85 | 0.24 | 0.88 |
| | MASDT | 0.32 | 0.89 | 0.35 | 0.79 | 0.33 | 0.74 |
| | AltFreezing | 0.29 | 0.89 | 0.48 | 0.84 | 0.43 | 0.56 |
| | *Average* | 0.27 | 0.82 | 0.35 | 0.84 | 0.34 | 0.78 |
| DAW-FDD (Ju et al., 2024) | EfficientNet-B3 | 0.21 | 0.88 | 0.30 | 0.89 | 0.39 | 0.91 |
| | RECCE | 0.24 | 0.84 | 0.41 | 0.92 | 0.20 | 0.87 |
| | EfficientNet-B4 | 0.21 | 0.84 | 0.28 | 0.87 | 0.54 | 0.90 |
| | MASDT | 0.28 | 0.92 | 0.46 | 0.83 | 0.17 | 0.86 |
| | AltFreezing | 0.21 | 0.92 | 0.39 | 0.87 | 0.37 | 0.79 |
| | *Average* | 0.23 | 0.88 | 0.37 | 0.88 | 0.33 | 0.87 |
| FairAlign (Ours) | EfficientNet-B3 | 0.17 | 0.90 | **0.11** | **0.97** | 0.23 | **0.93** |
| | RECCE | 0.12 | 0.94 | 0.32 | 0.93 | 0.20 | 0.90 |
| | EfficientNet-B4 | 0.12 | 0.76 | 0.29 | 0.87 | 0.39 | 0.92 |
| | MASDT | 0.18 | 0.93 | 0.37 | 0.83 | 0.18 | 0.85 |
| | AltFreezing | **0.11** | **0.94** | 0.29 | 0.89 | **0.15** | 0.89 |
| | *Average* | 0.14 | 0.89 | 0.28 | 0.90 | 0.23 | 0.90 |

across all three metrics. For skin tone-sensitive attributes, FairAlign achieves a state-of-the-art $G_{FPR}$ of 9.01, while the Baseline with the RECCE backbone and DAG-FDD with the same backbone attain slightly better $F_{FPR}$ and $F_{EO}$, respectively. Importantly, FairAlign achieves state-of-the-art detection performance in **all** scenarios, demonstrating that our method enhances fairness without compromising detection efficacy.

### A.3.3 Distributions of Gender and Skin tone in FF++ and CelebDF Datasets

We present the distributions for gender and skin tone sensitive attributes within the FF++ and CelebDF datasets in Fig. 4. In the FF++ dataset, a noticeable imbalance is evident in the distribution of skin tones compared to gender. Notably, skin tone 8 (Black) is significantly underrepresented, whereas skin tone 3 (White) is the most prevalent. The resulting imbalance factor—defined as the ratio of the highest to the scarcest category—is calculated to be **8.6**. In contrast, the gender distribution exhibits a more modest imbalance, with a male-to-female ratio of **1.8**. These findings suggest that biases associated with skin tone are likely to be more severe than those related to gender. A same pattern is observed for CelebDF dataset.

Table 9: DPR and DPD results on CelebDF. Best results in each column are in **bold** and second-best results are underlined.

| Methods | Backbones | Bias Metrics | | | | | |
| | | Gender | | Skin Tone | | Intersection | |
| | | DPD ↓ | DPR ↑ | DPD ↓ | DPR ↑ | DPD ↓ | DPR ↑ |
|---|---|---|---|---|---|---|---|
| Baseline | EfficientNet-B3 | 0.39 | 0.89 | 0.28 | 0.84 | 0.44 | 0.86 |
| | RECCE | 0.36 | 0.77 | 0.39 | 0.67 | 0.45 | 0.62 |
| | EfficientNet-B4 | 0.38 | 0.72 | 0.36 | 0.83 | 0.43 | 0.62 |
| | MASDT | 0.37 | 0.63 | 0.44 | 0.78 | 0.42 | 0.62 |
| | AltFreezing | 0.38 | 0.87 | 0.31 | 0.83 | 0.34 | 0.85 |
| | *Average* | 0.38 | 0.78 | 0.36 | 0.79 | 0.42 | 0.71 |
| DRO$_{\chi^2}$ (Hashimoto et al., 2018) | EfficientNet-B3 | 0.39 | 0.90 | 0.28 | 0.85 | 0.54 | 0.87 |
| | RECCE | 0.31 | 0.78 | 0.40 | 0.68 | 0.45 | 0.63 |
| | EfficientNet-B4 | 0.35 | 0.73 | 0.36 | 0.84 | 0.43 | 0.57 |
| | MASDT | 0.17 | 0.88 | 0.34 | 0.79 | 0.32 | 0.63 |
| | AltFreezing | 0.18 | 0.88 | 0.35 | 0.84 | 0.34 | 0.86 |
| | *Average* | 0.28 | 0.83 | 0.35 | 0.80 | 0.42 | 0.71 |
| MMD (Deka & Sutherland, 2022) | EfficientNet-B3 | 0.40 | 0.91 | 0.29 | 0.87 | 0.25 | 0.89 |
| | RECCE | 0.19 | 0.87 | 0.41 | 0.69 | 0.56 | 0.66 |
| | EfficientNet-B4 | 0.29 | 0.75 | 0.37 | 0.85 | 0.24 | 0.79 |
| | MASDT | 0.10 | 0.90 | 0.30 | 0.81 | 0.22 | 0.64 |
| | AltFreezing | 0.15 | 0.90 | 0.38 | 0.85 | 0.30 | 0.88 |
| | *Average* | 0.23 | 0.87 | 0.35 | 0.81 | 0.31 | 0.77 |
| DAG-FDD (Ju et al., 2024) | EfficientNet-B3 | 0.40 | 0.92 | 0.28 | 0.85 | 0.45 | 0.88 |
| | RECCE | 0.22 | 0.79 | 0.41 | 0.69 | 0.56 | 0.65 |
| | EfficientNet-B4 | 0.28 | 0.73 | 0.37 | 0.85 | 0.64 | 0.58 |
| | MASDT | 0.27 | 0.79 | 0.32 | 0.79 | 0.33 | 0.64 |
| | AltFreezing | 0.29 | 0.89 | 0.38 | 0.84 | 0.44 | 0.86 |
| | *Average* | 0.29 | 0.82 | 0.35 | 0.80 | 0.48 | 0.72 |
| DAW-FDD (Ju et al., 2024) | EfficientNet-B3 | 0.41 | 0.94 | 0.30 | 0.89 | 0.66 | 0.91 |
| | RECCE | 0.24 | 0.73 | 0.41 | 0.62 | 0.36 | 0.67 |
| | EfficientNet-B4 | 0.29 | 0.77 | 0.38 | 0.87 | 0.34 | 0.60 |
| | MASDT | 0.28 | 0.62 | 0.27 | 0.83 | 0.53 | 0.66 |
| | AltFreezing | 0.25 | 0.92 | 0.39 | 0.87 | 0.25 | 0.89 |
| | *Average* | 0.29 | 0.80 | 0.35 | 0.82 | 0.43 | 0.75 |
| FairAlign (Ours) | EfficientNet-B3 | 0.27 | **0.96** | 0.29 | **0.97** | 0.35 | **0.93** |
| | RECCE | 0.20 | 0.88 | 0.42 | 0.63 | 0.37 | 0.67 |
| | EfficientNet-B4 | 0.24 | 0.77 | 0.29 | 0.87 | **0.20** | 0.72 |
| | MASDT | **0.08** | 0.93 | **0.22** | 0.89 | 0.19 | 0.65 |
| | AltFreezing | 0.10 | 0.94 | 0.39 | 0.89 | 0.25 | 0.89 |
| | *Average* | 0.18 | 0.90 | 0.32 | 0.85 | 0.27 | 0.77 |

Table 10: Results on WildDeepfake dataset. Best results in each column are in **bold**.

| Methods | Backbones | Bias Metrics (%)↓ | | | | | | | | | Detection Metrics (%) | | | |
| | | Gender | | | Skin Tone | | | Intersection | | | Overall | | | |
| | | $G_{\text{FPR}}$ | $F_{\text{FPR}}$ | $F_{\text{EO}}$ | $G_{\text{FPR}}$ | $F_{\text{FPR}}$ | $F_{\text{EO}}$ | $G_{\text{FPR}}$ | $F_{\text{FPR}}$ | $F_{\text{EO}}$ | AUC↑ | FPR↓ | TPR↑ | ACC↑ |
|---|---|---|---|---|---|---|---|---|---|---|---|---|---|---|
| Baseline | RECCE | 5.39 | 5.39 | 8.38 | 10.09 | **11.19** | 14.63 | 13.54 | 39.59 | 66.95 | 64.31 | 20.25 | 83.21 | 83.25 |
| | EfficientNet-B4 | 3.45 | 3.45 | 5.17 | 14.86 | 16.14 | 21.32 | 17.04 | 38.71 | 66.42 | 67.39 | 21.19 | 87.24 | 89.67 |
| | *Average* | 4.42 | 4.42 | 6.77 | 12.47 | 13.66 | 17.97 | 15.29 | 39.15 | 66.68 | 65.85 | 20.72 | 85.22 | 86.46 |
| DAG-FDD | RECCE | 5.34 | 5.34 | 8.18 | 15.32 | 14.12 | **12.12** | 12.12 | 37.12 | 62.11 | 65.34 | 20.33 | 87.38 | 88.89 |
| (Ju et al., 2024) | EfficientNet-B4 | 4.32 | 4.32 | 6.38 | 16.09 | 22.45 | 24.21 | 19.21 | 40.75 | 69.98 | 65.46 | 22.09 | 85.33 | 89.16 |
| | *Average* | 4.83 | 4.83 | 7.28 | 15.70 | 18.28 | 18.16 | 15.66 | 38.93 | 66.04 | 65.40 | 21.21 | 86.35 | 89.02 |
| FairAlign | RECCE | 4.27 | 4.27 | 6.83 | **9.01** | 12.22 | 14.07 | **10.55** | **29.16** | **60.95** | 67.11 | 21.78 | 86.34 | 86.22 |
| (Ours) | EfficientNet-B4 | **1.15** | **1.15** | **3.13** | 12.83 | 17.11 | 19.29 | 15.01 | 35.78 | 63.49 | **69.42** | **19.11** | **88.09** | **90.11** |
| | *Average* | 2.71 | 2.71 | 4.98 | 10.92 | 14.66 | 16.68 | 12.78 | 32.47 | 62.22 | 68.26 | 20.44 | 87.21 | 88.16 |

Table 11: Results on FF++ dataset with ethnicity as the sensitive attribute.

| Backbone | Ethnicity | | |
| | $G_{FPR}$ | $F_{FPR}$ | $F_{EO}$ |
|---|---|---|---|
| **FairAlign** | | | |
| MASDT | 3.73 | 4.60 | 12.36 |
| AltFreezing | 3.84 | 4.67 | 12.69 |
| **Baseline** | | | |
| MASDT | 10.56 | 12.99 | 11.18 |
| AltFreezing | 8.11 | 7.21 | 21.48 |

### A.3.4 Performance on Ethnicity Sensitive Attribute

We also apply FairAlign on FF++ dataset with ethnicity as the sensitive attribute. We use MASDT and AltFreeing backbones. The results are shown in Table 11. The results clearly demonstrate that FairAlign significantly reduces ethnicity-based bias for both backbones. This confirms our method's effectiveness across all tested sensitive attributes (gender, skin tone, their intersection, and ethnicity).

### A.3.5 Impact of $\lambda$ on Fairness

In addition to the the impact of $\lambda$ on bias ($F_{\text{FPR}}$ Intersection) in the main paper, here we explore the relationship between and the other seven metrics for the CelebDF dataset versus $\lambda$ in Fig. 5. From this figure, we observe that for five out of the seven plots, $\lambda = 1$ results in the least amount of bias, while for $F_{\text{EO}}$ for Intersection and $G_{\text{FPR}}$ for Skin Tone, $\lambda = 0.25$ and $\lambda = 0.75$ are marginally better than 1. For consistency, we set $\lambda = 1$ throughout all the experiments in this work.

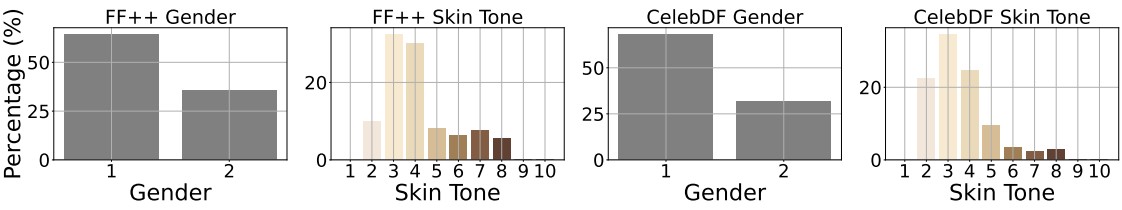

Figure 4: Distribution of genders and skin tones in the FF++ and CelebDF datasets.

(a) $F_{\mathrm{EO}}$ Skin Tone

(b) $F_{\mathrm{EO}}$ Gender

(c) $F_{\mathrm{EO}}$ Intersection

(d) $F_{\mathrm{FPR}}$ Skin Tone

(e) $G_{\mathrm{FPR}}$ Skin Tone

(f) $G_{\mathrm{FPR}}$ Gender

(g) $G_{\mathrm{FPR}}$ Intersection

Figure 5: Effect of tuning the $\lambda$ hyperparameter on bias metrics for AltFreezing backbone trained on CelebDF dataset.

