# OpenReview forum: "Kernel Space Conditional Distribution Alignment for Improving Group Fairness in Deepfake Detection"
_TMLR — Accepted by TMLR_

### Review · Reviewer_cYx4 · 2025-07-16

**Summary Of Contributions:**

The paper addresses the theme of fairness in deepfake detection. It proposes FairAlign, a method based on Maximum Mean Discrepancy (MMD) to quantify differences between the distributions of embeddings conditioned on different categories. This approach isolates bias and encourages the model to reduce it. The method is evaluated using the FF++, Celeb-DF, and WildDeepfake datasets.

**Audience:**

Yes

**Broader Impact Concerns:**

No concerns. The paper addresses a theme focused on alleviating various ethical biases in AI models.

**Claims And Evidence:**

No

**Requested Changes:**

1. Argue why skin tone in this quantized form is relevant for ethical AI (deepfake detection).

2. Explain the mathematical model in a clear and unambiguous way.

3. Explain why the results reported in Ju et al. are not replicable here.

4. Visual examples of skin tone ranges and prediction decisions, with detailed explanation, should be added to the supplementary material.

**Strengths And Weaknesses:**

On the **positive side**, the paper presents technical innovation, mathematical modeling, and broad evaluation (across multiple datasets and in comparison with recent, strong prior works).

On the **weakness side**:
1.  **Motivation – Skin Tone**
    - The paper emphasizes that it is one of  the first to address bias with respect to **skin tone** in deepfake detection. However, it fails to justify why skin tone is a relevant attribute for evaluating fairness in the context of ethical AI models.
    - In my view, **skin tone is of limited relevance**. "Ethnicity" is more directly tied to social biases, but skin tone is not necessarily indicative of ethnicity. *Image recorded skin tone* results from a non-linear combination of factors including ethnicity, tanning, image exposure, and white balance. Addressing bias related to exposure or white balance contributes marginally to the broader goal of ethical AI.
    - Furthermore, the paper quantizes skin tone into a 10-value discrete set (inferred from Figure 2 in the Appendix), but does not justify why these 10 categories are meaningful or ethically relevant. I argue that ranges 2 and 3, for instance, contain significant overlap in ethnicity, with differences likely due to exposure and white balance variations.
    - Additionally, skin tone is inherently a *continuous variable*, and the paper does not explain the need for quantization. The mathematical framework could support a conditional probability density function (PDF) without requiring discretization.

2.   **Method Clarity**
    The mathematical derivation of FairAlign lacks clarity and contains questionable assumptions:
      -  From the formalism, it appears that z represents the attribute. The paper restricts this to discrete categorical variables (e.g., gender: z ∈ {0,1}; skin tone: z ∈ {0,...,9}). Meanwhile, x represents the embedding—a continuous, high-dimensional representation of the input image. Given these differences, applying the same kernel function over z and x is unjustified.
      -  The method defines kernel functions $k_X$ and $k_Z$​ on the embedding and attribute spaces, respectively. Initially (" " Let’s define kernel functions $k_X$ and $k_Z$ on the space of embeddings X and Z, respectively"."), this suggests a mistake (treating z as an embedding). Later derivations, however, use kernels to project attributes into another space while still aiming for matching conditional PDFs. This raises a conceptual question: What is the meaning of fairness computed as differences in accuracy conditioned on a non-linear projection?
       - The paper does not specify what kind of kernel functions are used (e.g., RBF?), which is a critical detail.
        - The notation $\phi(x)=k_X(x,\cdot)$ is problematic. Since $k_X(x,\cdot)$ implies an inner product from a specific point $x_b$ to the key point, $x$​, using $\phi_x(x_b)$ would be more appropriate. Without specifying $x_b$​, $\phi(x)$ is not well-defined. Later notations with superscripts {a/b} are clearer, but then the introduction of both K and Φ/Ψ becomes redundant—they represent the same objects.
        - Equation (2) introduces four layers of abstraction and compact notation with minimal explanation. Subsequent notation builds on this, and to me, the left-hand side and right-hand side of the equation seem to have incompatible dimensions.
       -  Equation (7) is intuitive, but Equation (6) is not, due to unclear usage of kernels over categorical attributes and inconsistent notation.

3. **Evaluation Ambiguity**
    - A major concern is that results attributed to Ju et al. (2024) in this paper cannot be verified in the original work.
    -  While the paper introduces skin tone as a new attribute, evaluations with respect to gender and detection metrics should be reproducible.
    -  Ju et al. use an Xception backbone, while this paper uses different architectures. More worryingly, the results reported for Ju et al. and for the proposed method are significantly worse than those in the original paper.
    -  Later tables report results for EfficientNet-B3, and one would expect similar performance in gender and detection metrics. However, this paper shows noticeably worse outcomes.
    - The use of "Average" across selected models lacks justification. The selection of these particular models and the omission of the Xception backbone highlight inconsistencies and weaken the experimental design.

4.  **Supplementary Material**
   -  Although the main paper is limited to 12 pages, additional results can and have been included in the supplementary material.
   -  The paper fails to show visual examples of individuals categorized under different skin tone bins. Such examples are important to justify whether this attribute is meaningful and consistently annotated: each quanta should be easy to be linked to a ethnicity such that "comparable results" across values make sense
   -  It is also necessary to show multiple images within each skin tone bin to demonstrate variability and confirm that skin tone is a relevant factor in judging whether a deepfake detection model is biased.

---

> ### Author Response · Authors · 2025-08-26
>
> We thank the reviewer for their time and their valuable comments. Below we provide a careful point-by-point response to each comment. We would be happy to provide additional discussions/information if needed.
>
> > **Method is based on MMD**
>
> Kindly note that while we discuss MMD as a related domain adaptation technique, our method is in fact built upon the conditional kernel bures (CKB) metric and **not MMD**. This is a critical distinction, as the choice of the CKB metric is central to our contribution. As we argue in the paper, methods like MMD are effective in aligning marginal discrepancy but can inadvertently discard "valuable discriminating information in label distributions". In contrast, our approach utilizes the CKB metric, specifically designed for aligning conditional distributions.
>
> > **Why is skin tone a relevant sensitive attribute for evaluating bias in ethical AI**
>
> Using skin tone as a sensitive attribute is a well-established phenomenon in the fairness literature [1-2]; however, it is the first time we are investigating this matter in the context of *deepfake detection*. In the context of fairness in AI, sensitive attributes, like gender, skin tone, and ethnicity, are used to partition a population into groups for whom we want to ensure fair outcomes. **Several legal frameworks** explicitly list skin color as a sensitive attribute and prohibit discrimination based on it [3]. These frameworks include Equal Credit Opportunity Act [4], The Fair Housing Act, etc. Accordingly, we treat skin tone as the sensitive attribute and audit disparities with respect to it (and, where relevant, its intersection with gender). This choice is both legally grounded (protected status under anti-discrimination law) and methodologically standard in AI fairness literature.
>
>
> [1] Buolamwini, Joy, and Timnit Gebru. "Gender shades: Intersectional accuracy disparities in commercial gender classification." Conference on fairness, accountability and transparency. PMLR, 2018.
>
> [2] Mehrabi, Ninareh, et al. "A survey on bias and fairness in machine learning." ACM computing surveys (CSUR) 54.6 (2021): 1-35.
>
> [3] Chen, Jiahao, et al. "Fairness under unawareness: Assessing disparity when protected class is unobserved." Proceedings of the conference on fairness, accountability, and transparency. 2019.
>
> [4] 12 CFR Part 202 -- Equal Credit Opportunity Act (Regulation B). https://www.ecfr.gov/current/title-12/chapter-II/subchapter-A/part-202?toc=1
>
> [5] The Fair Housing Act. (2023, June 22). https://www.justice.gov/crt/fair-housing-act-1
>
> > **Why quantize the skin tone into 10-value discrete set; skin tone 2 and 3 may contain overlap in ethnicity; skin tone may be affected by illumination**
>
> Please kindly note that the 10-value discrete set is not a categorization developed for this paper. The Monk Skin Tone (MST) [1] scale is a 10-point scale developed to be more inclusive and representative than previous scales used in machine learning.
>
> Skin tone is an established sensitive attribute in AI fairness (please see our previous response). Additionally, skin tone and ethnicity are not the same characteristics [3]. Our work deliberately decouples the analysis of skin tone from ethnicity, as our central argument is that computer vision models react to direct visual signals. We focus on skin tone because "encoders are highly likely to discriminate based on color given its strong visual prominence". Therefore, the potential ethnic background of individuals within a specific skin tone category does not directly impact our research question.
>
>
> We also acknowledge the valid concern regarding environmental variations, like variations in illumination. While our goal in this paper is to show the implications of fairness with skin tone as the sensitive attribute in deepfake detection, in practice, special cameras and ambient lighting can be used to mitigate lighting conditions as best possible.
>
> [1] Skin tone Research at Google. https://skintone.google/the-scale
>
> [2] Schumann, Candice, et al. "Consensus and subjectivity of skin tone annotation for ML fairness." Advances in Neural Information Processing Systems 36 (2023): 30319-30348.
>
> [3] Telles, Edward, René D. Flores, and Fernando Urrea-Giraldo. "Pigmentocracies: Educational inequality, skin color and census ethnoracial identification in eight Latin American countries." Research in Social Stratification and Mobility 40 (2015): 39-58.

---

> ### Author Response · Authors · 2025-08-26
>
> > **Applying the same kernel function over z and x is unjustified**
>
> In our implementation, we use a Gaussian kernel for both the embedding space $\mathcal{X}$ and the attribute space $\mathcal{Z}$. Before applying the kernel, we first transform the discrete categorical attributes ($z$) into a numerical vector space using one-hot encoding. Applying a Gaussian kernel to vector representations of categorical data is a standard and mathematically sound technique in machine learning [1].
>
> [1] Duvenaud, David. Automatic model construction with Gaussian processes. Diss. 2014.
>
>
> > **Authors use kernels to project attributes into another space. What is the meaning of fairness computed as differences in accuracy conditioned on a non-linear projection?**
>
> We use two kernels: $K_X$ on the embedding space and $K_Z$ on the attribute space. The kernel on the attribute space is not turning attributes into embeddings we condition on. *It is only the standard RKHS tool to estimate conditional mean/covariance* as functions of $Z$. Conditioning remains w.r.t. the original variable $Z$, not w.r.t the non-linear projection of $\phi_Z(z)$. Additionally, All fairness metrics($G_{FPR}$/$F_{FPR}$/$F_{EO}$) are computed per original groups. The kernel never defines these groups.
>
> > **Clarification on kernel function**
>
> We thank the reviewer for pointing this out. In all the experiments, we employ Gaussian kernel for both the embedding space and the sensitive attribute space (this was reflected in the uploaded code). The sigma parameter in the Gaussian kernel is dynamically set to the mean of the pairwise squared Euclidean distances for each mini-batch. We now add this to the paper to make things clearer (page 8, Section "Implementation Details").
>
> > **The notation $\phi(x) = k_X(., x)$ is problematic and it is not well-defined**
>
> This notation is common in RKHS [1]. We want to further elaborate it here to clear any confusion.  We use the RKHS feature map $\phi_X:\mathcal{X}\\to\\mathcal{H}_X$ defined by
>
> $\phi_X(x) \=\ k_X(\cdot, x) \in \mathcal{H}_X.$
>
> Please kindly note that $k_X(\cdot, x)$ is not a single value or an inner product. The dot (.) is a placeholder for a variable, meaning that $k_X(., x)$ represents a function in $\mathcal{H}$. For any $x, x' \in \mathcal{X}$ the reproducing property holds:
>
> $<\phi_X(x), \phi_X(x') >_{\mathcal{H}} = k_X(x, x')$
>
> Similarly, for any function $f \in \mathcal{H}$ we have
>
> $f(x) = \langle f, \phi_X(x) \rangle_{\mathcal{H}_X}.$
>
> [1] Kolahdouzi, Mojtaba, and Ali Etemad. "Toward fair facial expression recognition with improved distribution alignment." Proceedings of the 25th International Conference on Multimodal Interaction. 2023.
>
> > **Equation 2 has dimension mismatch**
>
> Please kindly note that Equation (2) is the standard and well-established definition for the conditional covariance operator in a high-dimensional feature space [1]. Equation 2 essentially says the covariance of the embeddings X, given the attribute Z, is equal to the total covariance of X minus the part of the covariance that can be explained by Z. This is exactly analogous to how variance is partitioned in simpler statistical models.
>
>
> [1] Ren, Chuan-Xian, You-Wei Luo, and Dao-Qing Dai. "BuresNet: Conditional bures metric for transferable representation learning." IEEE Transactions on Pattern Analysis and Machine Intelligence 45.4 (2022): 4198-4213.
>
> > **Unclear usage of kernels over categorical attributes**
>
> Please see our response to the *Applying the same kernel function over z and x is unjustified* title.
>
> > **Explain why the results reported in Ju et al. are not replicable here.**
>
> Please kindly note that we re-trained the backbones using their methods inside our evaluation pipeline for an apples-to-apples comparison with FairAlign. There are significant differences between their setup and ours; As an example, they use stochastic gradient descent while we use AdamW. As another example, they use a fixed learning rate, while we decrease the learning rate during the training. Also, they use different method for face extraction. Under our protocol, some metrics improved relative to their reported numbers; for instance, on CelebDF we observe GFPR 3.56 vs. 3.81 (lower is better). For reproducibility, we have released our code and configuration files.
>
> > **Visual examples of skin tone ranges**
>
> We agree that showing the skin-tone bins helps readers to understand the method. We have added a panel of the 10-level Monk Skin Tone scale with representative face crops for each bin, alongside the corresponding color, in Appendix A.1, page 15.

---

### Review · Reviewer_bZWE · 2025-07-23

**Summary Of Contributions:**

This paper proposes a novel bias mitigation objective for deepfake detection. Furthermore, it develops a rigorous bias evaluation scheme for skin tone and examines fairness/utility trade-off in the deepfake detection realm.

**Audience:**

Yes

**Claims And Evidence:**

Yes

**Requested Changes:**

- It would be interesting to examine the impact of FairAlign for other sensitive attributes and the biases related to them, as existing works showed that reducing one kind of bias can exacerbate the others. For example, how the use of FairAlign for skin tone impact the gender-related biases?

- For the sensitivity analysis related to $\lambda$ (Figure 1), the utility measures should also be presented.

**Strengths And Weaknesses:**

**Strengths**
- The paper focuses on an important and contemporary research problem.
- The motivation and the differences from the existing work are presented clearly.
- In the fairness literature, MMD variants have been used for many applications. The paper employs a novel distance metric for conditional distribution alignment (Conditional Kernel Bures (CKB) metric), which is empirically shown to outperform MMD for deepfake detection.

**Weaknesses**
- In general, the paper uses existing strategies/metrics for a specific application (deepfake detection). Thus, its contributions are limited.
- Experimental settings can be improved (please see requested changes).

---

> ### Author Response · Authors · 2025-08-26
>
> We thank the reviewer for their time and their valuable comments. Below we provide a careful point-by-point response to each comment. We would be happy to provide additional discussions/information if needed.
>
> > **Limited contributions**
>
> We respectfully argue that our contribution is not merely the application of the existing methods, but a conceptual and empirical advancement for the field of fair deepfake detection. The novelty of our work is threefold:
>
> *1.* Our primary contribution is providing a key insight into the group fairness in deepfake detection. In the fairness enhancing methods, the goal is to make embeddings invariant to sensitive attributes. However, a straightforward alignment of embedding distributions between groups (e.g., via a standard MMD loss) can cause the loss of the label information. It risks removing not only the sensitive attribute information but also essential features needed for the deepfake detection task. By using the Conditional Kernel Bures metric, which operates on the conditional covariance operators of the embeddings and is designed for conditional distributions, our method is able to specifically target and remove the statistical dependency on the sensitive attribute, while better preserving the residual structure in the embeddings necessary for discriminating between real and fake content. **This is why our method offers better accuracy-fairness trade-off.**
>
> *2.* Our paper presents the first systematic method to analyze and mitigate skin tone bias in deepfake detection. This is a previously under-explored problem. Our contributions here are introducing a practical pipeline for annotating deepfake datasets with the Monk Skin Tone scale, and providing the first empirical evidence (Table 3) that skin tone is a distinct and more severe source of bias.
>
> *3.* We are the first to introduce a rigorous and quantitative analysis of the fairness-accuracy trade-off in the context of deepfake detection. While the metrics (Fairea, HM) are not new, their application to this problem is. This moves the evaluation of fair deepfake detectors beyond a simple, and often inconclusive, side-by-side comparison of fairness and accuracy scores. Our analysis (Table 5) provides a holistic and unified perspective on the models' fairness-accuracy trade-offs. This contribution provides a new standard for evaluation in this field.
>
> > **How the use of FairAlign for skin tone impacts the gender-related biases**
>
> The concern that mitigating one bias may exacerbate another is a critical issue in fairness research. To directly investigate this in the case of FairAlign, we conduct a new set of experiments to analyze the cross-attribute impact of FairAlign. Specifically, on the FF++ dataset with the MASDT and AltFreezing backbones, we trained three separate models:
>
> *1.* Baseline with no fairness-enhancing method.
>
> *2.* A model where FairAlign is applied to mitigate gender bias directly.
>
> *3.* A model where FairAlign is applied to mitigate skin tone bias.
>
> For all the models above, we report fairness metrics for gender as the sensitive attribute in the table below.
>
>
> | **Backbone**                 |    |       **Gender**       |             |
> | :--------------------------- | :----------: | :----------: | :---------: |
> |                              | $G_{FPR}$ | $F_{FPR}$ | $F_{EO}$ |
> |                     |      |   **Baseline**   |  |
> | MASDT              | 1.38        | 1.38        | 19.71        |
> |AltFreezing              | 2.82         | 2.82         | 10.54        |
> |    |  | **FairAling (Targeting Gender)** |  |
> | MASDT            | 0.29        | 0.29        | 12.00       |
> | AltFreezing                  | 1.74        | 1.74        | 6.03      |
> |    |  | **FairAling (Targeting Skin Tone)** |  |
> | MASDT            | 0.87        | 0.87        | 18.79       |
> | AltFreezing                  | 2.80        | 2.80        | 10.13      |
>
> The results show that even when FairAlign directly targets only skin tone, the gender bias is still reduced across all metrics compared to the baseline. For example, on AltFreezing, $F_{EO}$ improves from 10.54 to 10.13, and on MASDT, $G_{FPR}$ improves from 1.38 to 0.87. This demonstrates that FairAlign does not exacerbate the non-target bias beyond the baseline. Instead, it yields a positive, albeit smaller, fairness improvement on non-target attributes. We have added this discussion to the paper on page 11, Section "Cross-Attribute Impact".
>
> > **Adding utility measure to Figure. 1**
>
> Following the reviewer's comment, we have now updated Fig. 1 (Fig. 2 in the new version, page 9) in the paper. Now, Fig. 1 (Fig. 2 in the new version) shows both fairness and utility (AUC). This figure shows that FairAlign is capable of increasing AUC while reducing bias, and this is the reason why FairAlign offers a better fairness-utility trade-off.

---

> ### Comment · Action_Editor_7BPt · 2025-09-08
> **Can you submit your recommendation asap?**
>
> thanks.

---

### Review · Reviewer_hug7 · 2025-08-04

**Summary Of Contributions:**

1.	This paper proposes FairAlign, a new loss term for promoting group fairness for deepfake detection. This method is the loss function that uses the Conditional Kernel Bure (CKB) metric proposed in the paper (Lu&Ren, 2021). This method improves the group fairness of the state-of-the-art on the datasets FF++, CelebDF, and WildDeepfake.
2.	This paper also shows that FairAlign improves skin-tone fairness. The skin tones are classified by the MST scale (Heldreth et al, 2023b). This paper enhances fairness based on the intersection of gender and skin tone in the context of deep-fake detection for the first time.
3.	Two unified metrics are used for the first time in the realm of fair deep-fake detection. Fairea (Hort et al, 2021) and Harmonic Mean (Lesota et al, 2022: Li et al, 2023)are used to combine both fairness and accuracy into unified indices.

**Audience:**

Yes

**Claims And Evidence:**

No

**Requested Changes:**

- Clarify kernel function and compare with linear kernel to show the effectiveness of kernel space.
- Conduct feature embedding visualization, e.g., t-SNE, to support the claim of P1.(1).
- Conduct experiments on other sensitive attributes.
- P.4, the second paragraph of FairAlign, replace “Let’s” with “Let us”.
- P.5 Eq.(6), the third-fifth rows should be written in one row.
- (Heldreh 2023a) and (Heldresh 2023b) are the same article. They should be merged.
- P6, DPD, and DPR should be moved to the Appendix since they are reported only in the Appendix.

**Strengths And Weaknesses:**

Strengths
- Survey of Fairness research is well conducted, and the fairness evaluation measure is used.
- The performance of FairAlign is higher than that of the state-of-the-art method on fairness deep fake detection, DAW=FDD (Ju et al, 2024).
- Fairness accuracy -trade-off concerning skin tone is firstly improved in the research area of deepfake detection.
- The algorithm of face skin tone categorization is demonstrated, and code is provided.

Weaknesses

-  This paper applies existing methods for fairness in deep fake detection. Therefore, the novelty is limited to the aspect of machine learning methodologies.
CKB is an existing method. The authors only apply this method for improving Group Fairness in deepfake detection.
Though the authors introduce fairness-accuracy trade-off evaluation in deepfake detection, the introduced methods, Fairea and Harmonic Mean, are existing methods, which were used in different fairness problems.

-  It is not clear what kernel function is used. In addition, $z$ seems to be a label of a sensitive attribute. How is it treated?

-  P1. (1) of the second paragraph, the authors claim that ``the embeddings generated by these detectors contribute to retain information related to sensitive attributes, “but this fact is not supported.

-  Evaluation of other sensitive attributes, such as race and nationality, is lacking.

---

> ### Author Response · Authors · 2025-08-26
>
> We thank the reviewer for their time and their valuable comments. Below we provide a careful point-by-point response to each comment. We would be happy to provide additional discussions/information if needed.
>
> > **Novelty is limited to the aspect of machine learning methodologies**
>
> We respectfully argue that our contribution is not merely the application of the existing methods, but a conceptual and empirical advancement for the field of fair deepfake detection. The novelty of our work is threefold:
>
> *1.* Our primary contribution is providing a key insight into the group fairness in deepfake detection. In the fairness enhancing methods, the goal is to make embeddings invariant to sensitive attributes. However, a straightforward alignment of embedding distributions between groups (e.g., via a standard MMD loss) can cause the loss of the label information. It risks removing not only the sensitive attribute information but also essential features needed for the deepfake detection task. By using the Conditional Kernel Bures metric, which operates on the conditional covariance operators of the embeddings and is designed for conditional distributions, our method is able to specifically target and remove the statistical dependency on the sensitive attribute, while better preserving the residual structure in the embeddings necessary for discriminating between real and fake content. **This is why our method offers better accuracy-fairness trade-off.**
>
> *2.* Our paper presents the first systematic method to analyze and mitigate skin tone bias in deepfake detection. This is a previously under-explored problem. Our contributions here are introducing a practical pipeline for annotating deepfake datasets with the Monk Skin Tone scale, and providing the first empirical evidence (Table 3) that skin tone is a distinct and more severe source of bias.
>
> *3.* We are the first to introduce a rigorous and quantitative analysis of the fairness-accuracy trade-off in the context of deepfake detection. While the metrics (Fairea, HM) are not new, their application to this problem is. This moves the evaluation of fair deepfake detectors beyond a simple, and often inconclusive, side-by-side comparison of fairness and accuracy scores. Our analysis (Table 5) provides a holistic and unified perspective on the models' fairness-accuracy trade-offs. This contribution provides a new standard for evaluation in this field.
>
> > **Clarification on kernel function**
>
> We thank the reviewer for pointing this out. In all the experiments, we employ Gaussian kernel for both the embedding space and the sensitive attribute space (this was reflected in the uploaded code). The sigma parameter in the Gaussian kernel is dynamically set to the mean of the pairwise squared Euclidean distances for each mini-batch. We now add this to the paper to make things clearer (page 8, under implementation details).
>
>
> > **Comparison to linear kernel**
>
> To validate the effectiveness of employing a Gaussian kernel, we now conduct new experiments. In this study, we compare our proposed FairAlign with a Gaussian kernel against a variant using a standard linear kernel ($\mathcal{K}_{linear} (x_i, x_j) = x_i^tx_j$). We use FF++ dataset, treating gender and skin tone as the sensitive attributes. The results are presented below.
>
>
>
> | **Backbone**                 |  **Gender**  |              |             | **Skin tone** |              |             |
> | :--------------------------- | :----------: | :----------: | :---------: | :-----------: | :----------: | :---------: |
> |                              | $G_{FPR}$ | $F_{FPR}$ | $F_{EO}$ |  $G_{FPR}$ | $F_{FPR}$ | $F_{EO}$ |
> | &#8203;                      | &#8203;      | &#8203;      | **FairAlign (Gaussian Kernel)** | &#8203;       | &#8203;      | &#8203;     |
> | EfficientNet-B3              | 0.16         | 0.16         | 5.78        | 3.97          | 3.59         | 10.15        |
> | EfficientNet-B4              | 0.39         | 0.39         | 2.07        | 6.54          | 12.29         | 11.79        |
> | &#8203;                      | &#8203;      | &#8203;      | **Linear Kernel** | &#8203;   | &#8203;      | &#8203;     |
> | EfficientNet-B3            | 0.46        | 0.46        | 6.69       | 6.18         | 5.81        | 13.51       |
> | EfficientNet-B4                  | 0.48        | 0.48        | 2.75       | 10.42         | 13.78        | 16.64       |
>
>
>
> The results, presented in the table above, demonstrate a clear and consistent trend: the Gaussian kernel outperforms the linear kernel in reducing bias across both backbones and sensitive attributes. For instance, with the EfficientNet-B3 backbone and gender as the sensitive attribute, our proposed method with the Gaussian kernel achieves a $G_{FPR}$ of 0.16, a nearly three-times improvement over the linear kernel's score of 0.46. We have now added this experiment to the paper, on page 11, section Gaussian kernel effectiveness.

---

> ### Author Response · Authors · 2025-08-26
>
> > **T-sne for supporting the claim on page 1: the embeddings still retain information related to sensitive attributes**
>
> Following the reviewer's comment, we conducted a t-SNE visualization experiment to provide explicit empirical support for our claim that detector embeddings retain information about sensitive attributes. To this end, we extract embeddings from the final layer of an EfficientNet-B4 detector trained with the DAG-FDD bias mitigation method on the Celeb-DF dataset. We then project these embeddings into a 2D space using t-SNE and colored the resulting points based on the subjects' gender. The visualization reveals clear clustering patterns. Embeddings corresponding to male and female subjects occupy different regions in the 2D space. This empirically demonstrates that gender-related information persists in the embeddings. This is now added to page 1 and Figure 1 of the paper.
>
> > **Experiments on other sensitive attributes**
>
> Please kindly note that we already presented the results for the baseline method with **ethnicity** as the sensitive attribute in Table 3. Following the reviewer's comment, we also applied FairAlign on FF++ dataset, treating ethnicity as the sensitive attribute. The results are shown below.
>
> | **Backbone**                 |  **Ethnicity**  |              |             |
> | :--------------------------- | :----------: | :----------: | :---------: |
> |                              | $G_{FPR}$ | $F_{FPR}$ | $F_{EO}$ |
> |                     |      |   **FairAlign**   |  |
> | MASDT              | 3.73        | 4.60        | 12.36        |
> |AltFreezing              | 3.84         | 4.67         | 12.69        |
> |    |  | **Beseline** |  |
> | MASDT            | 10.56        | 12.99        | 11.18       |
> | AltFreezing                  | 8.11        | 7.21        | 21.48      |
>
> The results clearly demonstrate that FairAlign significantly reduces ethnicity-based bias for both backbones. This confirms our method's effectiveness across all tested sensitive attributes (gender, skin tone, their intersection, and ethnicity). This analysis is now added to the paper, on page 18, section "Performance on Ethnicity Sensitive Attribute".
>
>
> > **Typos and replacement of DPD and DPR definitions**
>
> We thank the reviewer for catching these typos. We have now corrected all the noted typos and performed a thorough proofreading to prevent recurrences in the final version. Also, following the reviewer's comment, we have now moved the definitions of DPD and DPR to the Appendix.

---

### Public Comment · ~Calvin_McCarter1 · 2025-07-14
**Conditional variants of MMD**

This is not at all a criticism of the FairAlign method, or of the paper overall, but it's worth noting that conditional variants of the MMD do exist. Such conditional variants have recently been used, for example, to measure the reliability of conditional sequence models [1] and to perform domain adaptation in the presence of confounding [2]. Having said that, I do think that the Conditional Kernel Bures metric used here is a superior choice, because it exploits the notion of transport cost to directly minimize the discrepancy between the conditional distributions.

[1] Glaser, P., Paul, S., Hummer, A. M., Deane, C., Marks, D. S., & Amin, A. N. (2024, July). Kernel-based evaluation of conditional biological sequence models. In Forty-first International Conference on Machine Learning.

[2] McCarter, C. Towards Backwards-Compatible Data with Confounded Domain Adaptation. (2024, November). Transactions on Machine Learning Research.

---

> ### Author Response · Authors · 2025-08-26
>
> Thank you for your thoughtful comment and for pointing us to this highly relevant and recent work on MMD variants. As truly noted by you, the main reason for proposing CKB is its superiority in aligning conditional distributions.
>
> We appreciate your constructive engagement with our research. Please let us know if you have further questions or concerns.

---

### Author Response · Authors · 2025-08-26

We sincerely thank the reviewers and the area chair for their time and constructive feedback. We are encouraged that the reviews recognized several key strengths of our work, including: (i) a well-designed and comprehensive fairness analysis, (ii) state-of-the-art performance, (iii) the first systematic investigation of skin tone bias in deepfake detection, and (iv) the use of novel distance metrics. We have carefully addressed all concerns raised in the individual response section. Below, we provide a concise summary of our responses and highlight the corresponding revisions to the manuscript. All changes in the paper are marked in **blue.**

- **Comparison to linear kernel:** We added a new experiment comparing our method against a linear kernel, demonstrating that our approach consistently outperforms it.

- **T-SNE visualization:** A new figure has been included using t-SNE to illustrate that embeddings indeed carry information about sensitive attributes.

- **Ethnicity as a sensitive attribute:** We extended our experiments by applying FairAlign to ethnicity, in addition to gender, skin tone, and their intersection.

- **Cross-attribute experiment:** We investigated whether mitigating bias with respect to skin tone could inadvertently exacerbate bias with respect to gender. Results show that FairAlign not only avoids worsening the bias compared to the baseline, but also yields a modest improvement in fairness.


- **Clarification of novelty:** We emphasized three main contributions of the paper: (i) achieving a superior fairness–accuracy trade-off, (ii) presenting the first analysis and mitigation of skin tone bias in deepfake detection, and (iii) quantifying the fairness–accuracy trade-off in this context.

- **Ethical justification of skin tone as a sensitive attribute:** We clarified that skin tone is a valid sensitive attribute with both established legal grounds and technical precedent.

- **Visualization of skin tone bins:** We added a figure depicting the Monk skin tone scale bins, along with representative face examples for each bin.

---

> ### Comment · Action_Editor_7BPt · 2025-08-27
> **The deadline has been extended for a week.**
>
> @authors, please indicate whether additional experimental results are pending.
> @reviewers, please read the author response and evaluate if and how it has changed your initial assessment of the manuscript.

---

> > ### Author Response · Authors · 2025-08-27
> >
> > Dear AC,
> >
> > Thank you for granting the one-week extension. We have completed all the requested experiments, and no further experiments are pending.

---

### Decision · Action_Editor_7BPt · 2025-09-19

**Recommendation:** Accept as is

**Additional Comments:**

The authors provided an anonymous code repo containing the implementation of experimental code.

**Audience:**

Yes

**Audience Explanation:**

Addressing the fairness-accuracy trade-off is a fundamental challenge in AI that is highly relevant for many application domains. As such, it is studied by several researchers in the TMLR community, who likely would show interest in this submission’s findings.

**Claims And Evidence:**

Yes

**Claims Explanation:**

Reviewers initially had concerns that the description of the proposed method was lacking in detail and somewhat ambiguous. It was also pointed out that the baseline is not particularly strong, as it did not match performance metrics in the corresponding publication. The authors addressed all concerns in their rebuttal, which the reviewers found largely satisfactory with some concerns remaining. Overall the authors support the claims stated in the manuscript appropriately.